# Singular Vectors of Attention Heads Align with Features

**Gabriel Franco** [1]  **Carson Loughridge** [1]  **Mark Crovella** [1,2]

## Abstract

Identifying feature representations in language models is a central task in mechanistic interpretability. Several recent studies have made the observation that feature representations can be inferred in some cases from singular vectors of attention matrices. However, sound justification for this phenomenon is lacking. In this paper we address that question, asking: why and when do singular vectors align with features? First, we demonstrate that singular vectors robustly align with features in a model where features can be directly observed. We then show theoretically that such alignment is expected under a range of conditions. We close by asking how, operationally, alignment may be recognized in real models where feature representations are not directly observable. We identify *sparse attention decomposition* as a testable prediction of alignment, and show evidence that it emerges in real models in a manner consistent with predictions. Together these results suggest that alignment of singular vectors with features can be a sound and theoretically justified basis for feature identification in language models.

## 1. Introduction

Improving the interpretability of language models is critical, e.g., to build a foundation for improving model safety (Anwar et al., 2024). A central task in interpretability is the identification of *feature representations* — the elucidation of how models represent the concepts that they manipulate.

There is now considerable evidence that many concepts in language (and other) models are represented as directions in one-dimensional or low-dimensional subspaces (Elhage et al., 2022; Olah et al., 2020; Mikolov et al., 2013; Alain & Bengio, 2018; Park et al., 2023; Gurnee & Tegmark, 2024; Gurnee et al., 2023; Marks et al., 2024; Levy & Geva, 2025; Engels et al., 2025; Kantamneni & Tegmark, 2025; Hernandez et al., 2024). This has been termed the *linear representation hypothesis* (LRH) (Park et al., 2023; Elhage et al., 2022). However, reliably identifying the actual linear representations used by models in arbitrary settings is still an unsolved problem. That is, the LRH suggests that model activations can be conceived as additive sums of features, but finding the proper decomposition of a particular activation into its constituent features is still an enormous challenge.

In this regard, a number of studies have examined the weights of attention heads, in particular decomposing those weights using singular value decomposition, as a framework for studying feature representations (Merullo et al., 2024; Ahmad et al., 2025; Pan et al., 2024; Franco & Crovella, 2024; 2025). These studies empirically demonstrate a surprising relationship: **features used by an attention head tend to be aligned with its singular vectors**.[1] This does not require each feature to be one-dimensional; for example, a feature may lie in a low-dimensional subspace spanned by a small set of singular vectors. But in fact, many features appear to be one-dimensional and approximately in the same direction as a single singular vector (Merullo et al., 2024; Franco & Crovella, 2024).

The alignment of singular vectors with features presents a natural and powerful tool for interpretability. This is because the singular vector basis of a head then provides a comprehensible and tractable search space for extracting features from model activations. Projecting a model activation onto the various subspaces defined by a head's singular vectors defines a discrete set of candidate features. These candidate features can then be analyzed in various ways, e.g., for causal impact on task performance.

Thus, the phenomenon of singular vector–feature alignment needs further exploration. Hence, our motivating question:

[1]Department of Computer Science, Boston University, Boston, USA [2]Faculty of Computing & Data Sciences, Boston University, Boston, USA. Correspondence to: Gabriel Franco <gvfranco@bu.edu>.

*Proceedings of the 43rd International Conference on Machine Learning*, Seoul, South Korea. PMLR 306, 2026. Copyright 2026 by the author(s).

---

[1]By singular vectors of an attention head, we mean the singular vectors of the head's QK matrix $\Omega = W_Q^\top W_K$. Details are introduced below.

**Research Questions**

When features are represented linearly, under what conditions will the singular vectors of attention head matrices align with those features? Can we understand singular vector-feature alignment theoretically, and find evidence of alignment in real models?

Note that there are really two surprises here. The first is that any singular vector aligns with any feature at all. But a second surprise is that, in order for multiple features to align with corresponding singular vectors, the features must be nearly orthogonal (since singular vectors are themselves orthogonal). A complete study necessitates understanding both phenomena: both the single-feature case and the multi-feature case. Together we refer to these two phenomena as *singular vector–feature (SVF) alignment*.

To explore these phenomena, we use a combination of toy-model studies and theory. We illustrate and explore SVF alignment using a toy model similar to (Elhage et al., 2022), which we extend to include an attention head. We then present theorems that confirm the observed SVF alignment, and establish conditions under which it provably occurs. We then use these observations to formulate testable predictions that are implied by SVF alignment, and confirm that those predictions hold in both toy and real models (GPT-2 and Pythia).[2]

We summarize our contributions as follows. Our base assumptions are that features are linearly represented and activations are formed by summing features. Then:

**Main Results**

- SVF alignment arises robustly in a setting where heads attend to specific feature pairs.
- The top singular vectors of a head align with the feature pair most important for computing attention.
- Features orthogonalize to minimize interference, allowing for SVF alignment across multiple feature pairs.
- SVF alignment implies a testable prediction called *sparse attention decomposition*, which we confirm arises in real models.

## 2. Background

The notion of *feature* has been given varying definitions in the literature. Here, we consider a feature to be a geometrically *and* semantically consistent representation that has some functional role in the model. Geometric consistency

is realized by treating features as vectors, and semantic consistency is realized by hypothesizing that attention heads compute attention values as a function of which features are present in tokens. This view encompasses both the geometric properties surfaced by linear probes and SAEs, as well as the semantic properties required by causal analysis.

SVF alignment refers to a situation in which a feature, represented as a vector, has a high cosine similarity to a singular vector of an attention head's QK matrix. We use $\Omega$ to denote the QK matrix, defined as $W_Q^\top W_K$, and the left and right singular vectors of $\Omega$ are the columns of $U$ and $V$ as given by the SVD of $\Omega$, ie, $\Omega = U\Sigma V^\top$.

SVF alignment has been empirically demonstrated in a number of recent studies. The authors in (Merullo et al., 2024) find that inter-layer communication in models can be understood as taking place in low-rank subspaces defined by the singular vectors of attention heads. Turning to (Ahmad et al., 2025), the authors show that a single attention head "can simultaneously implement multiple, independent computations" that "can be activated or suppressed via interventions along individual directions in the SVD basis." Next, the authors in (Pan et al., 2024) study the interaction between tokens (image patches) in vision transformers, and "propose that left and right singular vectors of the query-key interaction matrix can be seen as pairs of interacting feature directions," thereby implicitly equating singular vectors with features. Finally, our paper builds most directly on (Franco & Crovella, 2024; 2025) which also relied on an assumption of SVF alignment in order to trace circuits, and introduced the notion of sparse attention decomposition, which we also use here (in Section 5).

Although each of the above studies relies on the alignment of singular vectors with features, **no previous work has presented a detailed investigation of *why* and *when* SVF alignment occurs.** Finding initial answers to those questions is the goal of this paper.

Understanding why and when SVF alignment occurs is important. This is because when SVF alignment occurs, it offers a new strategy to attack a central problem in mechanistic interpretability: finding feature representations (Anwar et al., 2024; Sharkey et al., 2025). The question of how features are represented has driven a large body of work in mechanistic interpretability, with many studies showing that interpretable concepts in models are encoded in one-dimensional subspaces (Elhage et al., 2022; Olah et al., 2020; Mikolov et al., 2013; Alain & Bengio, 2018; Park et al., 2023; Gurnee & Tegmark, 2024; Gurnee et al., 2023; Marks et al., 2024) or in low-dimensional subspaces (Levy & Geva, 2025; Engels et al., 2025; Kantamneni & Tegmark, 2025; Hernandez et al., 2024). However, the strategies used to find feature representations to date all have drawbacks. The use of probing (Tenney et al., 2019; Hewitt

---

[2]Code to reproduce all results is available at: https://github.com/gaabrielfranco/svf-alignment

& Liang, 2019; Belinkov, 2022; Li et al., 2023; Marks & Tegmark, 2024) can identify information that is decodable from model activations, but this does not by itself establish that the corresponding feature representation is used by the model: probe performance depends on the probe class and dataset, can reflect memorization or the probe's own capacity, and requires careful baselines and controls (Hewitt & Liang, 2019; Belinkov, 2022). From a mechanistic interpretability perspective, the key limitation is that probing is primarily correlational, motivating recent work that supplements probes with activation interventions to test whether the identified directions causally affect model behavior (Li et al., 2023; Marks & Tegmark, 2024). An alternative strategy uses Sparse Autoencoders (SAEs) (Huben et al., 2024; Bricken et al., 2023), but these methods are expensive to train and have been shown to have significant drawbacks (Leask et al., 2025; Bushnaq; Gao et al., 2024; Bussmann et al.; Chanin et al., 2024); one reason for this may be that SAEs are built only from model activations and do not take into account model weights (bilalchughtai & Bushnaq).

In contrast to methods like SAEs, exploiting SVF alignment uses the relationship between model activations *and* model weights. When SVF alignment holds, one can in principle consider the singular vectors of an attention head to be 'candidate features' and then examine activations to see if those features are present, as in (Franco & Crovella, 2024; 2025). In practice, feature identification using SVF alignment simply boils down to decomposing activations in the SVD basis — which can be done on a per-prompt basis and in a single forward pass over the model. Compared to the use of SAEs or linear probes, this offers a much more straightforward, efficient, and scalable approach; and, as we show in the body of this paper, it has theoretical and empirical justification.

## 3. Methods

We will generally be considering the setting in which a head is computing attention over a set of *key* tokens $S = \{s_i\}_{i=1}^m$. Token $s_m$ is also the *query* token which we will denote as $r$.

Given a pair of tokens $(r, s)$ for some $s \in S$, we will say that a head 'attends to' $(r, s)$ when it computes a large attention value for $(r, s)$. There is no strict attention threshold that defines 'attending' to a token pair, but attention on the pair should at least be higher than the uniform distribution (i.e., $1/m$). As explained below, tokens are treated as sums of features. We assume that the head attends to a token pair because of the presence of certain features in the tokens. The set of features that can cause a head to attend to a token pair are the features 'of interest' to the head. We don't assume that all features are of interest to any given head — many features, even if present, will not affect a head's attention computation.

To explore the alignment of singular vectors with features, we use a toy model that allows both features and singular vectors to be directly observed. We start with the toy autoencoder model from (Elhage et al., 2022) defined over a universe of $N$ features $\{w_i \in \mathbb{R}^D\}_{i=1}^N$. An input to the model $f$ is a choice of *feature strengths*, constructed as $f_i = a_i b_i$ where $a_i \sim$ Bernoulli$(p)$ and $b_i \sim U(0, 1)$. Internally, the model represents inputs as vectors in $\mathbb{R}^D$; we refer to the internal representations as tokens (using language model terminology). The model constructs a token as $r = Wf$ where $W \in \mathbb{R}^{D \times N}$ is the matrix whose columns are the features $w_i$. The autoencoder seeks to reconstruct $f$ as $f' = \text{ReLU}(W^\top r + b)$ with $b \in \mathbb{R}^N$. The learned weights are $W$ and $b$ and the reconstruction loss is $\mathcal{L}_{\text{recon}} = \|f - f'\|_2^2$. In this model, ReLU and negative biases can eliminate some interference between representations, which lowers reconstruction loss. This model was extensively explored in (Elhage et al., 2022) and was shown to generate feature representations $\{w_i\}$ that "spread out" to occupy the space $\mathbb{R}^D$ approximately isotropically.

We extend this model to study the influence of the attention mechanism. We add an attention head that compares tokens $r$ and $s$ and generates an attention logit as $\ell = r^\top W_Q^\top W_K s$ for learned matrices $W_Q, W_K \in \mathbb{R}^{H \times D}$. $W_Q$ and $W_K$ always appear together, so as mentioned, we use $\Omega$ to denote $W_Q^\top W_K$. The head computes logits for a single query and multiple keys as $\ell_j(r, S) = r^\top \Omega s_j$, $j = 1, \ldots, m$. The head then computes an output $p_{\text{head}} = \text{Softmax}(\ell_1, \ldots, \ell_m)$. Figure 10 in Appendix A shows a diagram of the complete toy model.

To train the head, we assume that it should attend to a pair of tokens when specific pairs of features are present in the tokens. To allow for flexible experimentation, we parameterize the target at the logit level. For a pair of tokens $(r, s)$ denote $f^{(r)}$ and $f^{(s)}$ as the corresponding feature strengths. For the token pair $(r, s)$ we define the target logit as $\ell^T(r, s) = \sum_{ij} T_{ij} f_i^{(r)} f_j^{(s)}$. We then define $p_{\text{target}} = \text{Softmax}(\ell_1^T, \ldots, \ell_m^T)$ over the $m$ token pairs. This parameterization allows us to use $T_{ij}$ to specify desired attention patterns in an intuitive fashion. We can interpret $T_{ij}$ as the amount that should be added to the target logit to the extent features $w_i$ and $w_j$ are present in the corresponding tokens.

The training loss for the head is defined as $\mathcal{L}_{\text{attn}} = \text{Cross-Entropy}(p_{\text{head}}, p_{\text{target}})$. The overall loss for the model is $\mathcal{L} = \mathcal{L}_{\text{recon}} + \lambda \mathcal{L}_{\text{attn}}$. Details of the model and parameter sweeps showing robustness of results are provided in Appendix A.

**Why this model?** The idea that tokens are sums of features is consistent with much of the work in model interpretability. It is strongly supported by the residual-connection structure

of the model as explained in (Elhage et al., 2021). It does not depend on the LRH but is fully consistent with the LRH. Similarly, much work in model interpretability assumes that heads attend to tokens because of the presence of specific features in those tokens. Hence the setting we describe here reflects only a minimal, commonly-adopted view of model internals. At the same time, our toy model allows for simultaneous learning of both feature representations ($W$) and model weights ($\Omega$) and direct comparison of the two.

## 4. Singular Vectors Align with Features

To start, we examine a model incorporating $N = 20$ features, each represented in $D = 10$ dimensions; the head dimension is $H = 10$. As a warm-up we look at how features $w_i$ (columns of $W$) arrange when the attention head is *not* included in the toy model. Cosine similarities among the features are shown in Figure 1(a). This setting is similar to the settings studied in (Elhage et al., 2022), and we see similar results: features arrange themselves into 10 pairs, one per dimension, with each pair antipodal. This arrangement is *isotropic* — features are spread equally in all directions. Formally, isotropy is the condition that $WW^\top$ is a multiple of the identity (ie, $W$ is a tight frame whose feature components are uncorrelated). As explained in (Elhage et al., 2022), this arrangement is expected as it minimizes *interference* — the reconstruction error caused by non-orthogonality of features. More details showing isotropy of features are in Appendix C.1 (Figure 19).

**Single-Feature-Pair Alignment.** Next, we observe the effects of adding the attention head. We study the simplest possible case: the head attends to a token pair when feature $w_0$ is present in the query token and feature $w_1$ is present in the key token. To implement this we set $T_{01} = 1$, and $T_{ij} = 0$ elsewhere. We train the model, decompose the head's weight matrix into its SVD $\Omega = U\Sigma V^\top$, and examine the alignment between features and singular vectors. Figure 2(a) shows that after training, **singular vectors are aligned with features.** Feature $w_0$ is aligned with left singular vector $u_0$, and feature $w_1$ is aligned with right singular vector $v_0$. The spectrum of $\Omega$ shows only a single large singular value, meaning that the head has allocated one of its dimensions entirely to representing and recognizing this feature pair.

*Why does SVF alignment happen?* Intuitively, the need for $w_0^\top \Omega w_1$ to output a relatively large value tends to align the vector $\Omega w_1$ with $w_0$. At the same time, the need for $w_i^\top \Omega w_1$ to output a smaller value for $i \neq 0$ means that $\Omega w_1$ is less attracted to each of the other $w_i$s. The resulting tendency for $\Omega w_1$ to be in the same direction as $w_0$, and likewise $w_0^\top \Omega$ to be in the same direction as $w_1$, means that $\Omega^\top \Omega w_1 \approx \alpha w_1$, ie, $w_1$ is approximately a right singular vector of $\Omega$. An analogous argument establishes that $w_0$ is approximately a

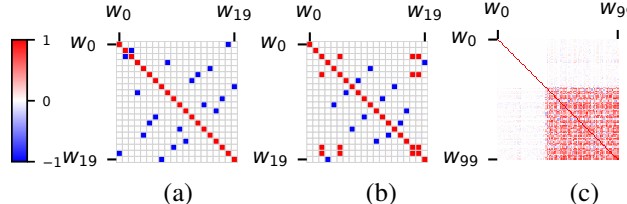

*Figure 1.* The geometry of features as illustrated via cosine similarities. (a) Without the attention head, features arrange isotropically. (b) With 20 features of which $w_0, w_1$ are of interest, features of interest orthogonalize against the others. (c) With 100 features in dimension 50, and 40 of those features are of interest (20 pairs), features of interest also orthogonalize against the others.

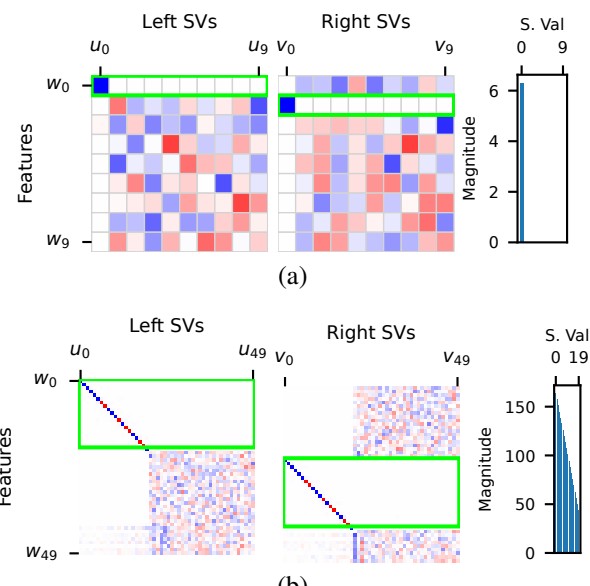

*Figure 2.* Singular vectors align with features (shown by green boxes). Cosine similarities of singular vectors and features, and magnitudes of singular values. (a) 20 Features of which $w_0, w_1$ are of interest; $w_0$ aligns only with $u_0$ and $w_1$ aligns only with $v_0$. (b) 100 features of which $w_0 \ldots w_{39}$ are of interest. For clarity, only an initial subset of features is shown; full figures are in Appendix A.

left singular vector of $\Omega$.

We make this argument precise by showing that SVF alignment *provably* occurs under fairly general conditions. For analysis, we allow the sets of features in the two tokens to differ, so columns of $X$ are features appearing the query token and columns of $Y$ are features appearing in the key tokens. Define the Gram matrices $\Sigma_X = XX^\top, \Sigma_Y = YY^\top$, and denote the value of $\Omega$ after training as $\Omega^\star$. As in our simulations, we assume training using Softmax computed over target logits that depend on feature presence.

**Theorem 1.** *(Informal) Assume the head's target logits are such that $\ell^T(r, s) = 1$ iff $x_1$ is present in $r$ and $y_1$ is present in $s$, and $0$ otherwise. Then after training, $\Omega^\star$ is rank-1, with left and right singular vectors $u_1 \propto \Sigma_X^{-1} x_1$ and $v_1 \propto \Sigma_Y^{-1} y_1$.*

Thus, *regardless of correlation among features,* the top singular vectors of $\Omega^\star$ are given by the covariance-whitened features $x_1$ and $y_1$. (See Appendix B.2 for the precise statement and the proof.) The case of isotropic features then immediately follows:

**Corollary 1.** *In the same setting as Theorem 1, if feature sets $X$ and $Y$ are in isotropic position, the singular vectors $u_1$ and $v_1$ will be exactly aligned with $x_1$ and $y_1$.*

*Proof.* If features are in isotropic position, $XX^\top \propto I$, $YY^\top \propto I$, and the result follows from Theorem 1. □

Further, we show in Theorem 2 (Appendix B.2) that even when features deviate from isotropy, singular vector alignment occurs approximately if the interference (inner products) between the features is sufficiently bounded.

Turning to the geometry of features, Figure 1(b) shows a second effect. Features $w_0$ and $w_1$ are orthogonal to *all other features.* This is not necessary for alignment per se, and deserves investigation. Intuitively, the remaining features have shifted into the $D - 2$ dimensional subspace that is orthogonal to the span of $\{w_0, w_1\}$.

We hypothesize that orthogonalization occurs to minimize reconstruction loss. This is provably the case for a model like ours. Theorem 3 in Appendix B.3 considers the case in which $\Omega$ is fixed while features are allowed to vary. It shows that in a setting where there is a penalty for interference among features (as in our toy model), the solution found to the training objective of the model will be the one in which features are orthogonal.

**Multi-Feature-Pair Alignment.** SVF alignment extends to cases involving multiple feature pairs. We observe that when multiple feature pairs are of interest to the head,[3] all features are typically still aligned with singular vectors. As an example, we show a model run with 100 features and hidden dimension 50. We set the target logits for feature pairs as $T_{i,i+20} = 26 - i$ for $0 \leq i < 20$ (zero otherwise). This causes 20 feature pairs $(w_i, w_{i+20})$ to be attended by the head, with logit values linearly declining in $i$. Figure 2(b) shows that all features align with singular vectors, and that the singular values of the head reflect the values of the target logits.

We attribute the phenomenon of alignment in the multi-feature case to the interaction of the two effects: alignment of the top singular vectors to most important features, and the resulting orthogonalization of the remaining features. Figure 1(c) shows the geometric arrangement of features in this case, confirming orthogonalization of the remaining features. We hypothesize that during training, singular vectors align with features to minimize attention loss, while at the

---

[3]For a number of features up to the head dimension; we discuss below.

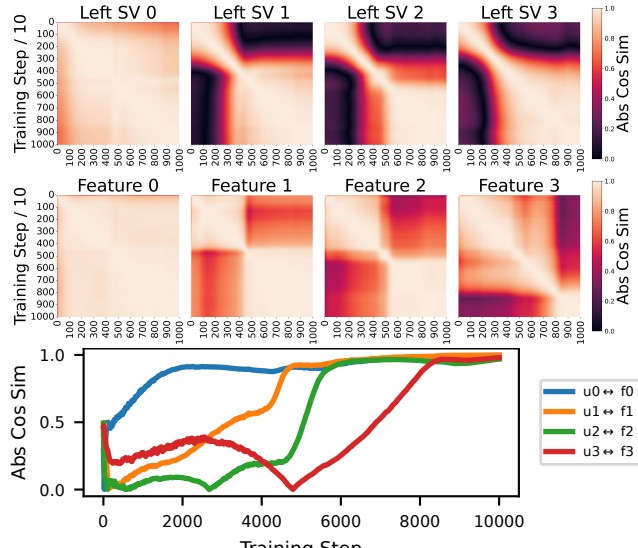

*Figure 3.* Both singular vectors and features evolve during training, and alignment occurs for highest-logit features first. Above: Cosine similarities showing evolution of singular vectors (top) and features (middle). Below: Cosine similarities showing evolution of alignment of singular vectors with features.

same time features orthogonalize to minimize reconstruction loss. Because singular vectors can shift during training, features can orthogonalize without impacting attention loss.

Evidence for this hypothetical mechanism comes from observation of the evolution of features and singular vectors during training. We use as an example a model run with 20 features and 10 hidden dimensions. We again set linearly declining target logits, in this case for only four pairs of features $(w_i, w_{i+4})$ for $i = 0, 1, 2, 3$. We confirm at the end of 10,000 training steps that features have aligned with singular vectors.

The dynamics of SVF alignment during training are shown in Figure 3, which compares vectors across training. Singular vectors align with features individually over time, in order of their impact on attention loss. The middle heatmap shows that feature $w_0$ changes little over training, while the top heatmap shows that left singular vector $u_0$ moves into its final position around step 1500. The bottom plot shows that this movement brings singular vector $u_0$ into alignment with feature $w_0$, matching the behavior formalized by Theorem 1 and Corollary 1. The next mode follows later: the top heatmap shows left singular vector $u_1$ shifting over the period up to about step 4000, after which the middle heatmap shows feature $w_1$ abruptly shifting. This is consistent with Theorem 3 in Appendix B.3: to reduce reconstruction interference, feature $w_1$ shifts toward a position orthogonal to feature $w_0$. Thus, both singular vectors and features evolve in tandem, with the pattern repeating for the remaining features and singular vectors.

The results shown here are robust over a wide range of model

parameters. In Appendix A we show that SVF alignment arises consistently over variations in relative loss weight $\lambda$, number of features $N$, context length $m$, head dimension $H$, and across random seeds.

**Alignment under Anisotropy.** So far, we have used isotropy as a simplifying assumption to show how Theorem 1 applies to the toy model via Corollary 1. However, we emphasize that the presence of feature anisotropy does not necessarily destroy SVF alignment. To illustrate, we first assess the range of anisotropy found in a real model (GPT-2), using SAE dictionary elements as a proxy for features; we find that $\|E_X\|_2$ (the metric used in Theorem 2) ranges between 10 and 55 depending on the layer (Appendix C.2). Next, we run an experiment based on a standard configuration of the toy model (Appendix C.3). We adjust the toy model to freeze the features and only train the head weights. Then, starting from isotropic features, we introduce controlled anisotropy into the feature set varying over the range of values found in GPT-2. Figure 4 shows that even when anisotropy gets large (close to the maximum seen in GPT-2), SVF alignment is good, with mean cosine similarity above 0.75. We also use Figure 4 to illustrate that (a) Theorem 2 is a lower bound that can be quite loose in practice; and (b) as suggested by Theorem 1, anti-whitening singular vectors using $\Sigma_X$ can significantly improve SVF alignment.

The above results have not taken into account the effects of rotary positional encoding (RoPE) (Su et al., 2024). In Appendix D we extend the toy model to include RoPE and show that SVF alignment can still occur, both for features whose target logits are position-independent and for features with position-dependent logits.

Finally, we note that we have so far studied the setting in which the number of features of interest to the head is less than the head capacity $H$. We believe that studying this regime is itself important, and can provide a foundation for follow-on studies of the regime in which there are more features of interest than head dimensions. In Appendix E we use the toy model to probe the latter regime, noting that some features move into superposition within the head's representation space – but for the majority of singular vectors, SVF alignment still holds.

---

**Results: Toy Model**

- The presence of an attention head causes singular vectors to align with features.
- Multiple features will align with corresponding singular vectors, up to the dimension of the head.
- Both features and singular vectors evolve during training to come into alignment.

---

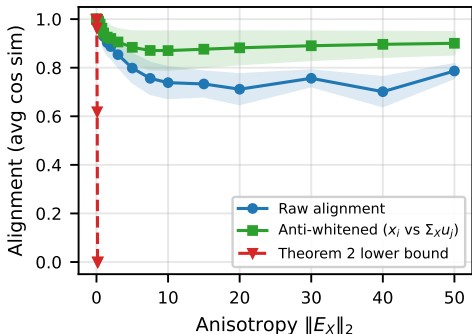

*Figure 4.* SVF alignment under anisotropic features. Features are varied over a range corresponding to observed anisotropy in GPT-2.

## 5. Sparse Attention Decomposition

The preceding sections provide theoretical and empirical support for the hypothesis that under some conditions, the singular vectors of an attention head will be aligned with features of interest to the head. The next question is: **does SVF alignment happen in real models?** This is important, because if it does happen, then by examining singular vectors in a model we may be able to identify actual features being used by that model. In fact, this strategy was taken in (Merullo et al., 2024; Ahmad et al., 2025; Pan et al., 2024; Franco & Crovella, 2024; 2025).

In a real model, we cannot conclusively establish that a singular vector is aligned with a particular feature without *a priori* knowledge of that feature's representation in the model. However, the hypothesis that singular vectors and features are aligned does lead to predictions that are testable in a real model. We focus on one prediction: *sparse attention decomposition* (Franco & Crovella, 2024; 2025). We first analyze logits to build intuition, and then we extend the analysis to attention.

**Analyzing Logits.** To illustrate, suppose features are aligned with singular vectors, either exactly (as in Corollary 1 or Figure 2) or approximately (as in Theorem 2 or Figure 4). Further, features that are not of interest are (approximately) orthogonal to features that are of interest, as in Figures 1(b) or 1(c) or Theorem 3. We'll term this set of assumptions the *alignment hypothesis*.

Now consider a head's logit decomposed in the singular vectors of $\Omega$:

$$\ell(r, s) = r^\top \Omega s = \sum_k r^\top u_k \sigma_k v_k^\top s \qquad (1)$$

$$= \sum_k \sum_{i,j} f_i^{(r)} \underbrace{w_i^\top u_k}\ \sigma_k \underbrace{v_k^\top w_j}\ f_j^{(s)}. \qquad (2)$$

The bracketed terms show that we can think of the attention

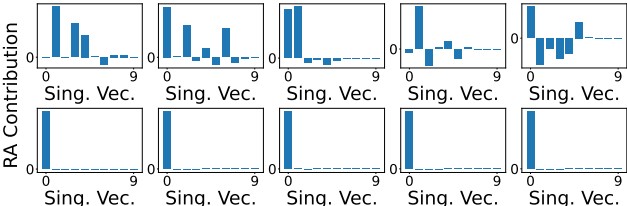

*Figure 5.* Relative attention decomposition is sparse when a single feature pair is present. Top: Early in training; Bottom: Late in training.

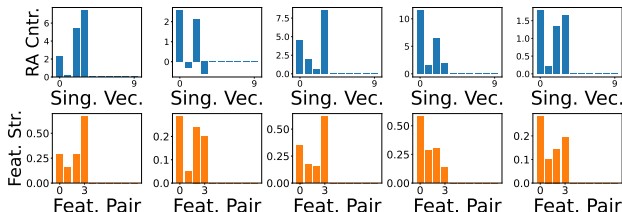

*Figure 6.* Sparse attention decomposition identifies feature presence. Top: Decomposition of relative attention across all 10 singular vectors for five token pairs $(r, s)$. Bottom: Feature strength $(f_i^{(r)} f_{i+4}^{(s)})$ for the four corresponding feature pairs $w_i, w_{i+4}$.

head as 'testing' each feature against each singular vector, and outputting a large value only when two features match a common pair of singular vectors (ie, having the same $k$). Under the alignment hypothesis, an individual term $(k, i, j)$ in (2) will only be large if features $i$ and $j$ are present, and aligned with singular vectors $k$. Now, in a real model we cannot in general observe the terms in (2) – however, we *can* observe the terms in (1). Because terms in (2) will only be large for singular vectors (values of $k$) aligned with features present in the tokens, the same is true of (1). In other words, when a head assigns a large logit $\ell$ to a pair of tokens because they contain a specific, limited set of features, the logit decomposed in the SVD basis can show a sparse representation.[4]

Sparse attention decomposition provides a powerful tool for interpretability, for two reasons. First, the presence of a few large terms in (1) suggest that the corresponding singular vectors may encode *representations* of features. And second, if attention is sparsely decomposed in the SVD basis, then only a small set of dimensions in the tokens contain features important for the attention computation. This 'reduces the search space' of causal features, a concept that is important in (Merullo et al., 2024; Franco & Crovella, 2024; 2025).

**Extending to Attention.** Next we consider how computing attention via Softmax affects sparse attention decomposition. Placing attention on a token pair requires putting a higher logit on the attended token than on the other key tokens. Hence when a model attends to $(r, s_j)$, the alignment hypothesis would predict not that $\ell(r, s_j)$ *per se* is sparse in the SVD basis, but rather that $\ell(r, s_j) - \ell(r, s_i)$, $j \neq i$, is sparse in the SVD basis.

To extend this observation to an arbitrary context length $m$, we use the notion of *relative attention* (Franco & Crovella, 2025). Over a set of logits $\ell_j(r, S)_{j=1}^m$, relative attention on token $j$ is defined as $\tilde{\ell}_j = \ell_j - \frac{1}{m-1} \sum_{i \neq j} \ell_i$. As shown in (Franco & Crovella, 2025) this metric has the useful property that if attention on $(r, s_j)$ is greater than $1/m$ (the uniform distribution) then $\tilde{\ell}_j > 0$. To decompose relative attention

we can simply apply (1) with $s = s_j - \frac{1}{m-1} \sum_{i \neq j} s_i$.

**Evidence in the Toy Model.** First, we show that sparse attention decomposition emerges during training. We illustrate using the model run from Figure 3 in which there are four feature pairs of interest to the head. Figure 5 shows the decomposition of relative attention for inputs where feature pair $(w_0, w_4)$ are the only features of interest present, both early and late in training. At first, relative attention does not show sparse decomposition, but later on the single feature of interest results in only one significant contribution to relative attention. In Appendix F we show additional examples (Figure 24) showing that sparsity does *not* emerge when features of interest are *not* present.

As Figure 5 suggests, a contribution from a particular singular vector pair is indicative of the presence of a corresponding feature pair in the inputs. In Figure 6 we show further confirmation of this effect. The figure shows that the strength of contribution to relative attention correlates with feature strength in the input. This supports the use of SVF alignment as a means of identifying the features of interest that are present in a token pair.

We show emergence of sparsity quantitatively using the metric from (Rolls & Tovee, 1995): $S(v) = (\frac{1}{n} \sum_i |v_i|)^2 / \frac{1}{n} \sum_i v_i^2$. This metric takes on a value of 1 when inputs are minimally sparse (all equal), and a value of $1/n$ when inputs are maximally sparse (only one nonzero value). For any given head and token pair, we compute $S(v)$ over the decomposition of relative attention.

Figure 7(a) shows how sparsity emerges in the toy model. We consider three classes of token pairs: pairs where only one feature of interest is present, pairs where two are present, and pairs where no features of interest are present; we measure decomposition sparsity using the $S(v)$ metric. The figure shows that when features of interest are present, attention decomposition is sparse, and it is sparser when fewer features are present. On the other hand, when features of interest are absent, attention decomposition is much less sparse. The dynamics are consistent with the evolution of SVF alignment seen in Figure 3, supporting the notion that

---

[4]The observed sparsity is *not* attributable simply to the attention matrix being low-rank or ill-conditioned, as we discuss below.

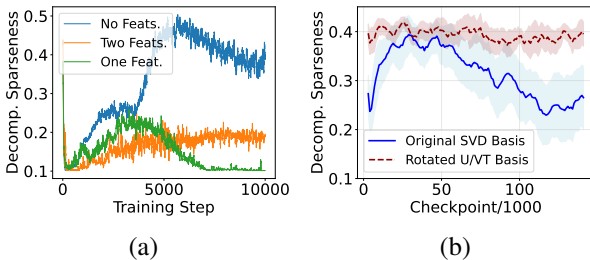

(a)                                    (b)

*Figure 7.* Sparse attention decomposition emerges during training: $S(v)$ metric, smaller is sparser, shaded regions are 95% confidence intervals. (a) Toy Model (b) Pythia-160M, for the IOI attention heads and token pairs identified in (Tigges et al., 2024). Sparsity is not due to presence of a small number of large singular value, as shown by the comparison to spectrum-preserving rotations of the SVD bases.

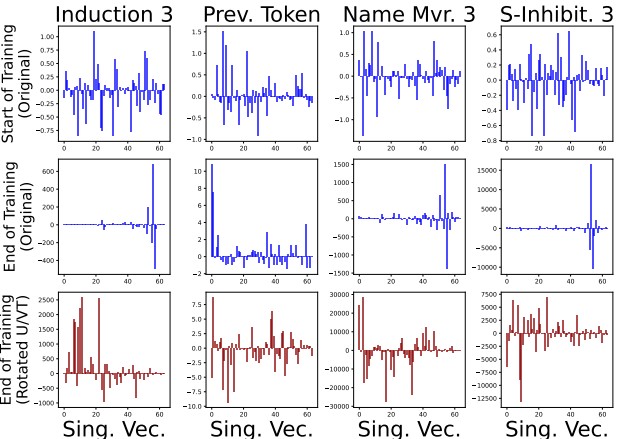

*Figure 8.* Sparse attention decomposition emerges after training in Pythia. Top: Early in training; Middle: Late in training. Bottom: Sparse decomposition does not arise with randomly rotated singular vectors. Note that singular vectors are ordered by decreasing singular value, showing that often the largest contributors to relative attention come from among the *smallest* singular values.

as singular vectors align with features, attention decomposition becomes sparse.

**Evidence in Language Models.** As discussed, sparse attention decomposition is a prediction of SVF alignment that is testable in real language models. We start with Pythia-160M (Biderman et al., 2023), which provides 130 checkpoints taken at intervals of 1000 training epochs. We input to the model a prompt from the Indirect Object Identification (IOI) task, and we capture relative attention at the heads and token pairs previously identified as part of the IOI circuit (Tigges et al., 2024). Further details are in Appendix G.

Figure 8 shows the decomposition of relative attention for four attention heads, at the start and end of training. In each case, relative attention decomposition becomes significantly sparse, with most of the contribution to relative attention coming from a small set of singular vectors of the

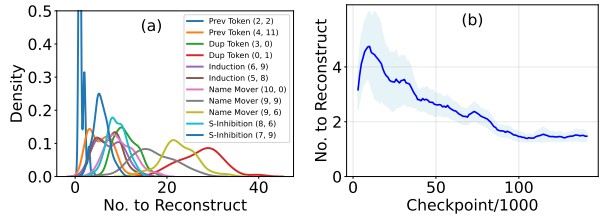

*Figure 9.* No. of singular vectors to reconstruct relative attention. (a) GPT-2, after training; (b) Pythia, during training, 95% CI.

attention head. Not all attention heads show such strong sparsity emergence (we present all head decompositions in Appendix G), which may be due to a larger number of features playing a role in some attention heads.

Quantitatively, attention sparsity emerges in Pythia in a manner similar to the dynamics of the toy model. Figure 7(b) shows the evolution of the $S(v)$ metric during training of Pythia. Here, $S(v)$ is averaged over all heads in the IOI circuit, and the general shape of sparsity emergence is similar to the single-feature case for the toy model.

Sparse attention decomposition is *not* attributable simply to the presence of a few large singular values in attention matrices. First, Figure 8 (middle) shows that the largest terms in (1) can be those with the *smallest* singular values, which is the opposite of what would be expected if low-rank properties were the cause. More directly, we show that sparse decomposition is not a result of a few large singular values by recomputing $S(v)$ after randomly rotating the $U$ and $V$ singular vector matrices. This modification reorients singular vectors *without* changing the matrix spectrum or rank. Figure 7(b) shows that the decline in $S(v)$ disappears when the SVD basis is randomly rotated, showing that the observed sparsity is due to the specific alignment of certain directions with certain singular vectors. Figure 8 (bottom) shows visually the absence of sparsity under rotated $U$ and $V$.

As suggested by Figure 6, the contributions made by singular vectors to relative attention can indicate the presence of features of interest in the input. As a measure of the potential number of features of interest present in a token pair, we define $N_{\text{recon}}(j)$ to count the minimum number of singular vectors needed to 'reconstruct' relative attention on key token $s_j$.[5] This metric separates the relative attention contributions of singular vectors into 'signal' and 'noise.'

Using $N_{\text{recon}}$ we can estimate how many features of interest may be present when an attention head attends to a token pair. Figure 9(b) shows how this metric changes over the duration of training in Pythia. It shows that for the heads considered here, typically only 1-4 singular vectors are needed

---

[5]Specifically, $N_{\text{recon}}(j)$ is the size of the smallest set of terms in (1) whose sum equals or exceeds the relative attention on key token $s_j$.

to reconstruct relative attention and shows how this value declines during training (consistent with Figure 7(b)).

As further confirmation of sparse attention decomposition, we examine the $N_{\text{recon}}$ metric for GPT-2 on the IOI task (Wang et al., 2023). Figure 9(a) averages over 128 IOI prompts that vary names, prompt templates (including name order), and objects. It shows distributions of the reconstruction count for different attention heads of the trained model. Typical values vary reflecting different functional roles of the attention heads, but they show that in most cases, $N_{\text{recon}}$ is small, showing that features of interest occupy low-dimensional subspaces, and suggesting that only a limited number of features are of interest to each head.

Finally, we note that SAD is present in both toy and real-world models. Within our real-world experiments, we evaluate two distinct architectures to mitigate the risk of architectural dependency in our results: Pythia, which uses RoPE, and GPT-2, which does not.

> **Results: Sparse Attention Decomposition (SAD)**
>
> - Under SVF alignment, if a limited number of features of interest are present, attention will decompose sparsely in the SVD basis (SAD).
> - In both toy and real models, we see SAD.
> - In real models, SAD evolves during training in a manner consistent with the toy model.

## 6. Discussion

**Limitations.** In this paper, directly examining features in real models is out of scope. However, a number of studies have shown in real models that features derived from SVF alignment are interpretable (Ahmad et al., 2025; Pan et al., 2024; Franco & Crovella, 2024) and causal for model performance (Franco & Crovella, 2025). Hence our focus here has instead been on clearly establishing the underlying theoretical and empirical evidence for SVF alignment.

While the evidence shows that singular vectors align closely with features in the toy model, it is possible that in real models at least some singular vectors align with 'cone directions' (directions that are overrepresented among features) due to anisotropy (Godey et al., 2024; Li et al., 2025). Our preliminary study (Appendix C.3) shows that SVF alignment can still occur when features are strongly anisotropic. Further, the fact that singular-vector derived features are often interpretable (Ahmad et al., 2025; Pan et al., 2024; Franco & Crovella, 2024) suggests, at minimum, that many singular vectors are *not* influenced by 'cone directions,' but more work is needed to understand the importance of feature anisotropy on SVF alignment.

While our results shed light on why and when the singu-

lar vectors of attention heads align with features, we only scratch the surface of the phenomenon. For example, there are questions related to the *number* of features that are of interest to a head. When there are more features of interest to a head than the head has dimensions to represent ($H$ in our model), how are features deconflicted? We present some evidence in Appendix E that the least 'important' features will share a common singular vector pair, but more work is needed, both experimentally and theoretically, to understand what can happen in general. Second, our work is limited to the study of a single head. Across the multiple heads in a real model, how are singular vectors in aggregate allocated among features? Note that there is evidence that an important benefit of multi-head attention is the ability to deconflict features, ie, to suppress noise in the attention mechanism arising from feature interference (Adler, 2025). Further, there is evidence that some singular vectors are nearly identical across multiple heads (the so-called 'control signals' in (Franco & Crovella, 2025)). Third, is there a higher level at which to view the set of singular vectors in use across all heads in a model? The authors in (Jermyn et al., 2023) show a toy-model case in which multiple heads work in superposition to compute a single output. Can we observe coordinated use of singular vectors across multiple heads?

## 7. Conclusions

Identifying features in language models is a central and open problem. We have shown empirical and analytical evidence that the singular vectors of an attention head will tend to align with features of interest to the head, when features are represented linearly. SVF alignment therefore provides a conceptual bridge between attention head weights and the features used by the model. Thus, and despite a number of open questions our study leaves, our results show that SVF alignment has potential as a valuable tool for feature identification in language models.

**Acknowledgments.** This work benefited from early feedback from Aaron Mueller and Micah Adler. The authors gratefully acknowledge consultations with ChatGPT in developing the theorems and Claude Code in developing the experiments. This research was funded by a grant from Coefficient Giving and by NSF award CNS-2312711.

## Impact Statement

This paper presents work whose goal is to advance the field of Machine Learning. There are many potential societal consequences of our work, none of which we feel must be specifically highlighted here.

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

# A. Robustness of SVF Alignment in Toy Model

## A.1. Toy Model Details

As described in the body of the paper, our model builds on the toy model used in (Elhage et al., 2022). We add to that model an attention head as described in Section 3. Figure 10 shows the complete toy-model setup.

The model is optimized using AdamW. Learning rate begins at 1e-3, and follows a cosine decay. The model uses a batch size of 1,024 key tokens, and 1,024 / $m$ query tokens. We train until results stabilize, which takes between 10,000 and 80,000 steps depending on the configuration.

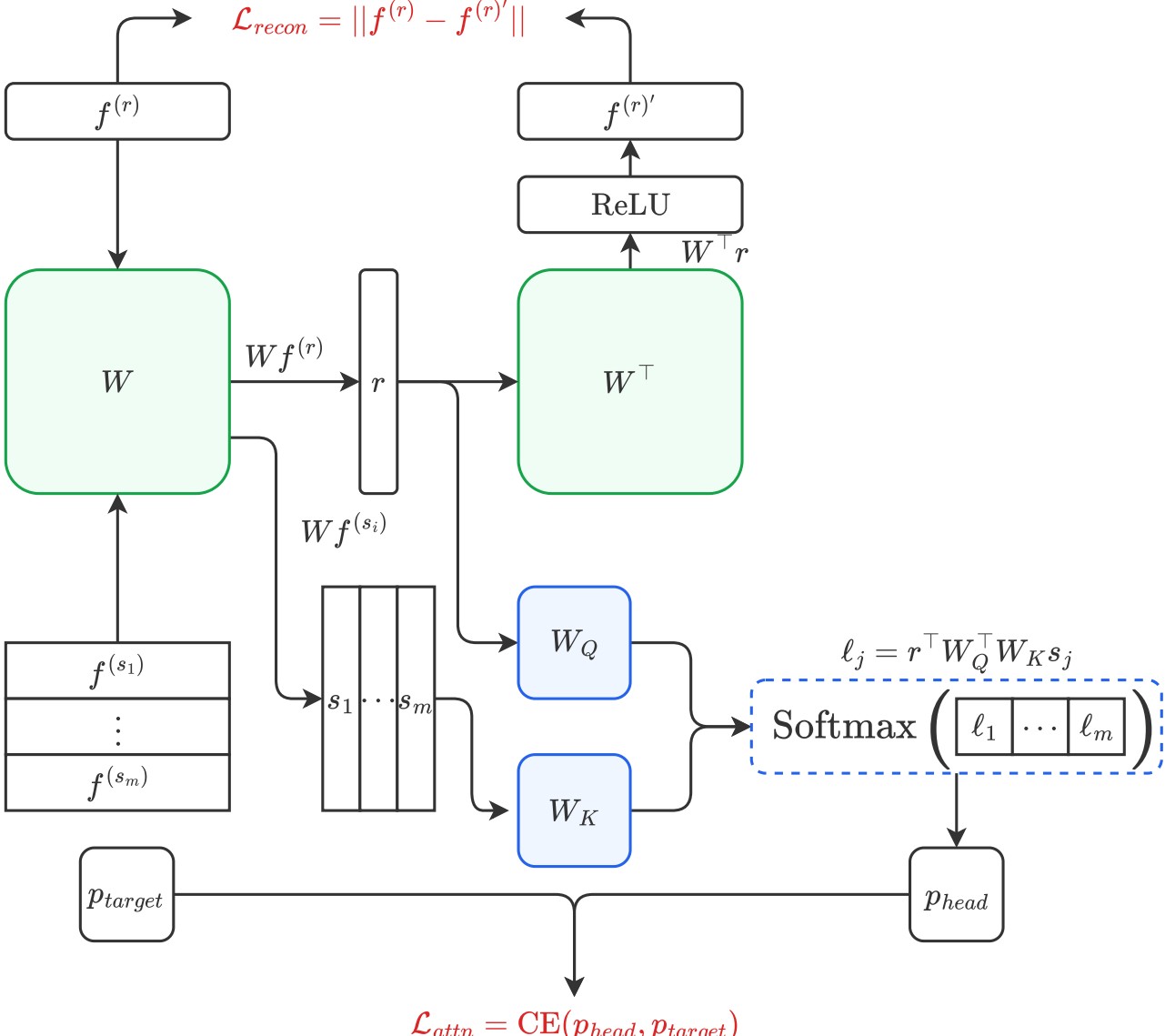

*Figure 10.* Complete schematic of the toy model used in Section 3. The model converts feature-strength vectors $f$ to tokens $Wf$, applies an attention head to query-key token pairs, and trains with reconstruction and attention losses.

**Full Alignment Plots** Here we show complete plots of the alignment of singular vectors with features for the two cases we discuss in Section 4. Figure 11 shows that singular vectors align with features in both cases.

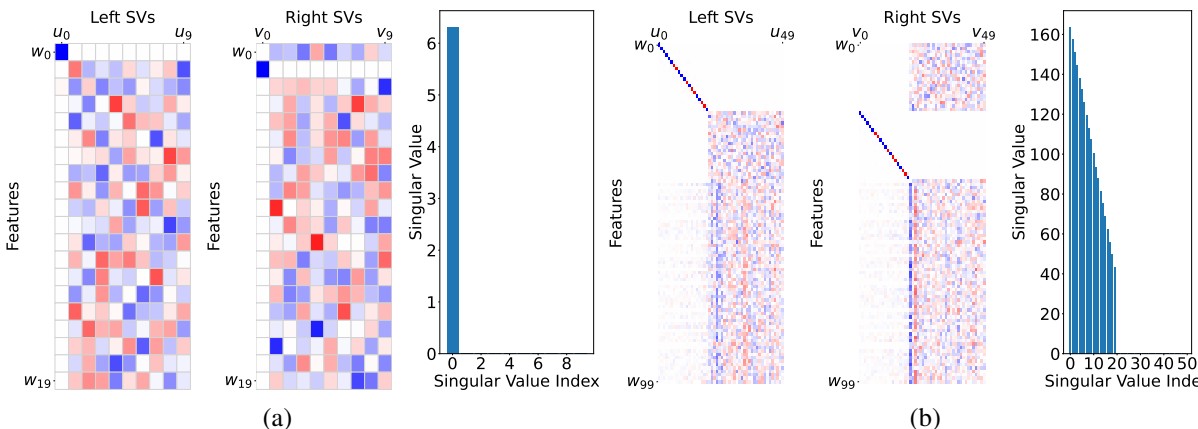

(a)                                                                                            (b)

*Figure 11.* Singular vectors align with features. For $\Omega = U\Sigma V^\top$, cosine similarities of $W$ and $U$, $W$ and $V$, and spectrum (diagonal of $\Sigma$). This is an expanded view of Figure 2. (a) 20 Features of which $w_0, w_1$ are of interest; $w_0$ aligns with $u_0$ and $w_1$ aligns with $v_0$. (b) 100 Features of which $w_0 \dots w_{39}$ are of interest. Due to sign ambiguities in SVD, SVF alignment yields cosine similarities that are either both positive or both negative.

## A.2. Robustness

Here we show that SVF alignment arises robustly across a wide range of model parameters. We show this by starting from a default model configuration and varying various aspects of the model. The default configuration is shown in Table 1.

| | | |
|---|---|---|
| $N$ | Number of features | 20 |
| $D$ | Token dimension | 10 |
| $H$ | Head dimension | 10 |
| $\lambda$ | Weight of $\mathcal{L}_{\text{attn}}$ in loss | 4 |
| $m$ | Context length | 4 |
| $p$ | Feature probability | 0.52 |

*Table 1.* Default Model Settings for Parameter Sweeps

Throughout this section, we study the case in which four feature pairs are of interest. Feature pair $(0, 4)$ has target logit of 24, feature pair $(1, 5)$ has target logit of 21, feature pair $(2, 6)$ has target logit of 18, and feature pair $(3, 7)$ has target logit of 15. These values are chosen to provide sufficient difference in attention after application of Softmax. As discussed in Section 4, the feature pair having the largest logit will generally map to the first singular vector, etc. To demonstrate SVF alignment we report the absolute cosine similarity between a feature and the singular vector that it corresponds to, in light of this mapping.

**Feature Probability.** As described in Section 3, features are present in the token with probability $p$. We study feature probabilities from 1 down to 0.02 in exponentially declining steps. Note that for feature probabilities greater than 0.05 in the default case, the token will in expectation be the sum of multiple features. Thus for most cases we study, the attention head does not 'see' any features themselves; features are mixed together in a token.

Figure 12 shows that under default settings, alignment occurs robustly across a range of feature sparsities, down to 0.038. At the level of 0.038 or less, feature pairs do not occur often enough in the input for the model to optimize for them. For example, at the level of $p = 0.02$, a given feature pair will only occur in less than 1% of inputs. Figure 13 demonstrates that alignment occurs more readily during training when features are present more often.

**Loss Weight $\lambda$.** The hyperparameter $\lambda$ is the weight of $\mathcal{L}_{\text{attn}}$ in the model's loss function and so controls the relative importance of reconstruction loss versus attention loss in training the model. Figure 14 shows that SVF alignment is robust over a range of $\lambda$ values spanning two orders of magnitude.

**Number of Features.** There are three dimensional parameters in the model: the number of features $N$, the hidden dimension $D$, and the dimension of the attention head $H$. We study a range of settings by holding the hidden dimension fixed and varying the other two.

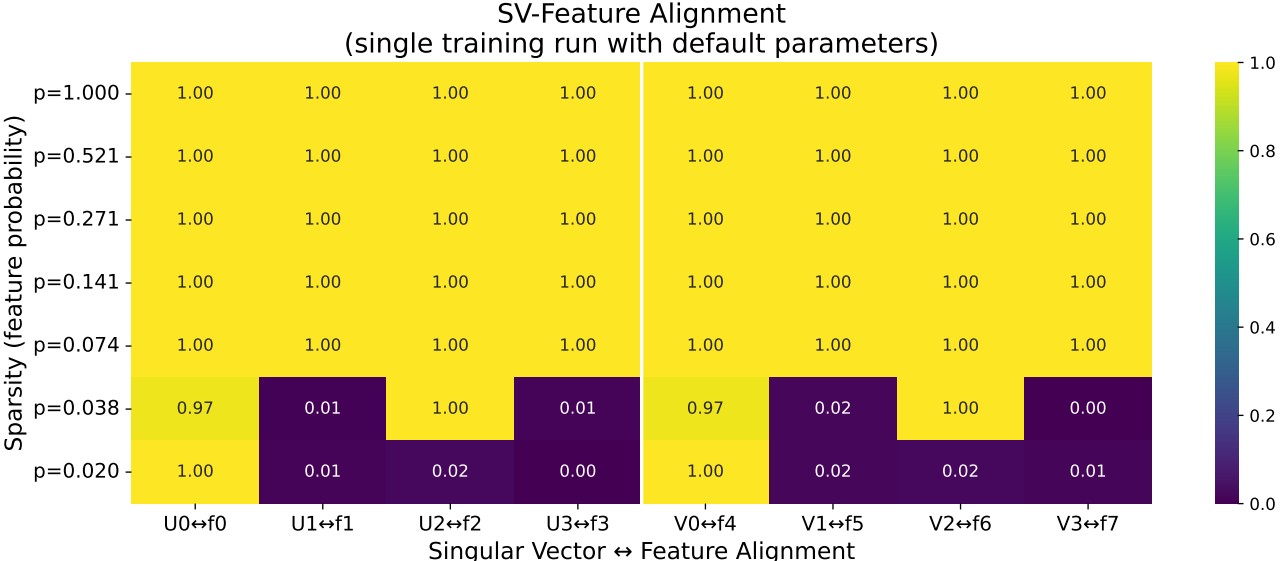

*Figure 12.* SVF Alignment is robust to feature probability.

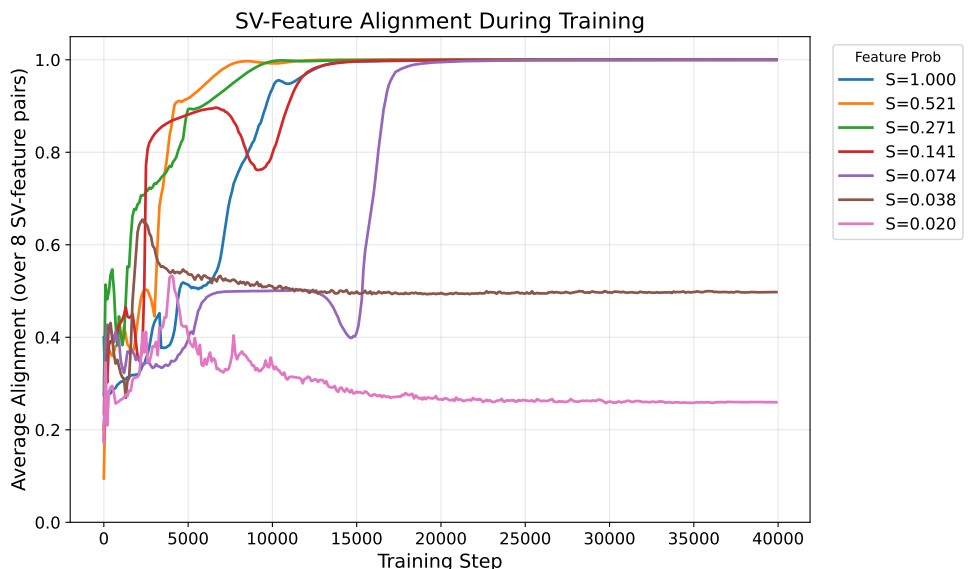

*Figure 13.* SVF Alignment arises earlier when features occur more frequently

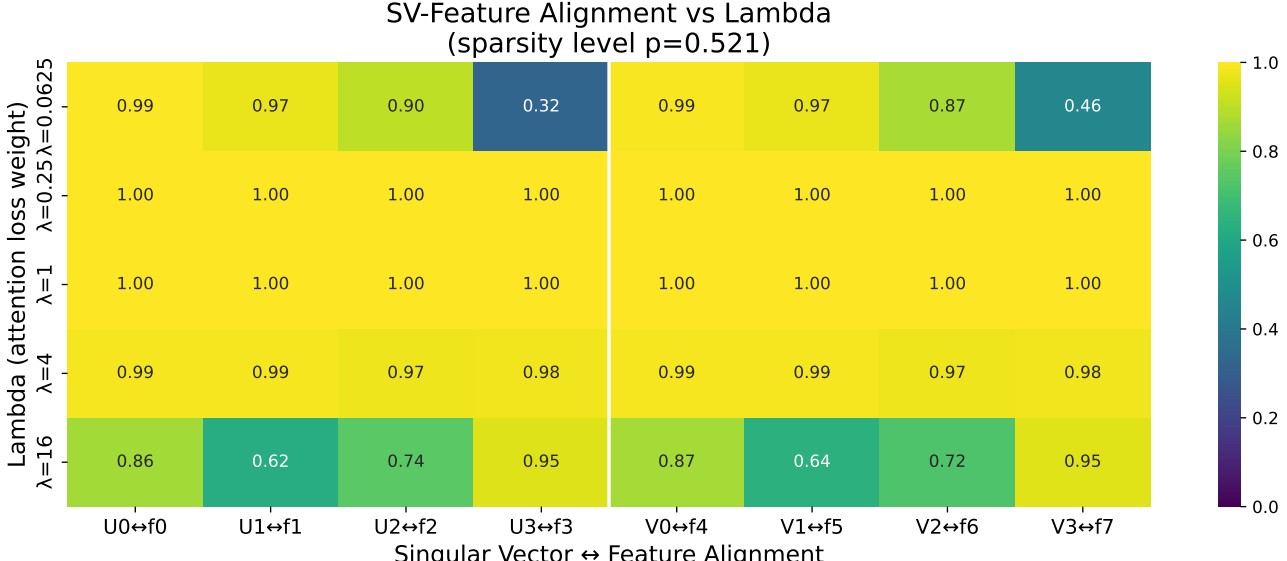

*Figure 14.* SVF Alignment is robust to $\lambda$.

First we show that when the number of features varies from the size of the hidden dimension up to three times its size, SVF alignment robustly occurs. Figure 15 shows SVF alignment for $N = 10, 15, 20, 25, 30$.

**Head Dimension.** Next we show that our results are consistent across varying dimension of the attention head. The head's dimension must be less than or equal to the hidden dimension, so we study configurations in which head dimension is 10, 8, 6, and 4. Figure 16 shows SVF alignment occurs robustly here as well.

**Context Length.** The model applies Softmax to logits over a context of size $m$. Figure 17 shows that SVF alignment occurs over a range of context lengths.

**Random Seeds.** Finally, in Figure 18 we show that SVF alignment reliably occurs over 5 random seeds.

## B. Theoretical Results

### B.1. Roadmap to the Theorems

In this section we present theorems that support the experimental results in Section 4.

The first set of theorems (1 - 2) show alignment of singular vectors when features are fixed, attention weights $\Omega$ are allowed to vary, and a single feature pair $(x_1, y_1)$ is of interest to the head.

**Theorem 1** shows that $\Omega$ will be rank-1, and it shows the form that the singular vectors take, which depends on $(x_1, y_1)$ and the covariance of features.

**Corollary 1** shows that if features are in isotropic position (also called a tight frame), then the top singular vectors of $\Omega$ will exactly align with the features $(x_1, y_1)$.

**Theorem 2** shows that if the features deviate from isotropic position by a small amount, the top singular vectors will still be directionally close to $(x_1, y_1)$.

Theorem 3 considers the alternative case: attention weights $\Omega$ are fixed, features are allowed to vary. Here, there is a reconstruction loss penalty on the features (as in our model).

**Theorem 3** shows that if two feature pairs are of interest to the model, and the first feature pair is already aligned with singular vectors of $\Omega$, the second feature pair will be induced to be orthogonal to the first feature pair.

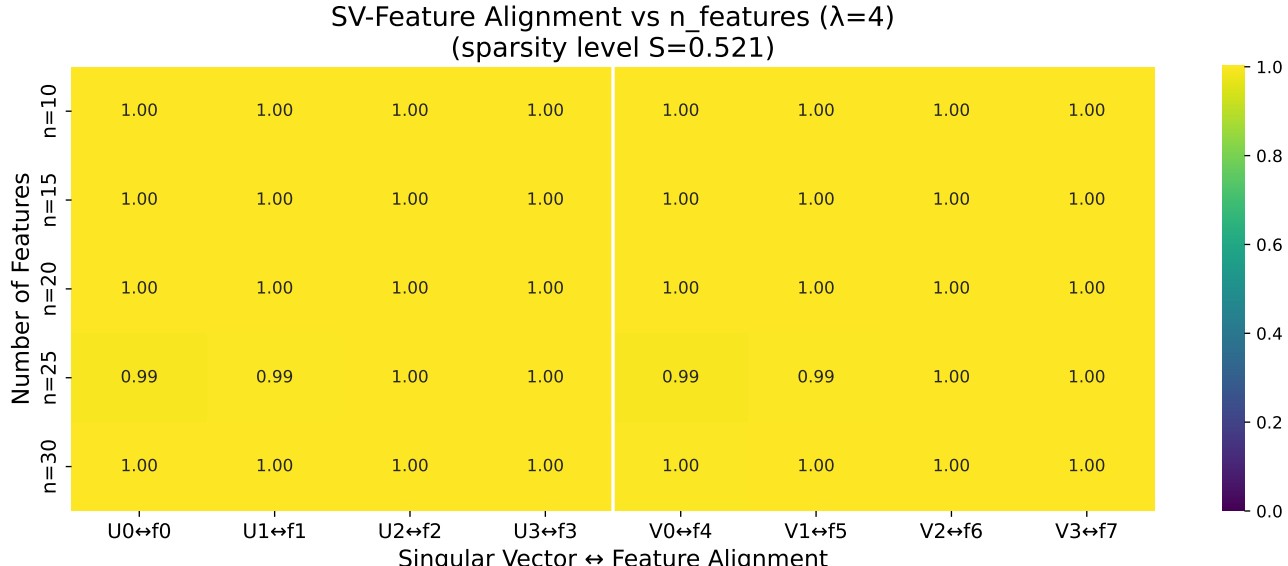

*Figure 15.* SVF Alignment is robust to varying number of features (compared to $D = 10$).

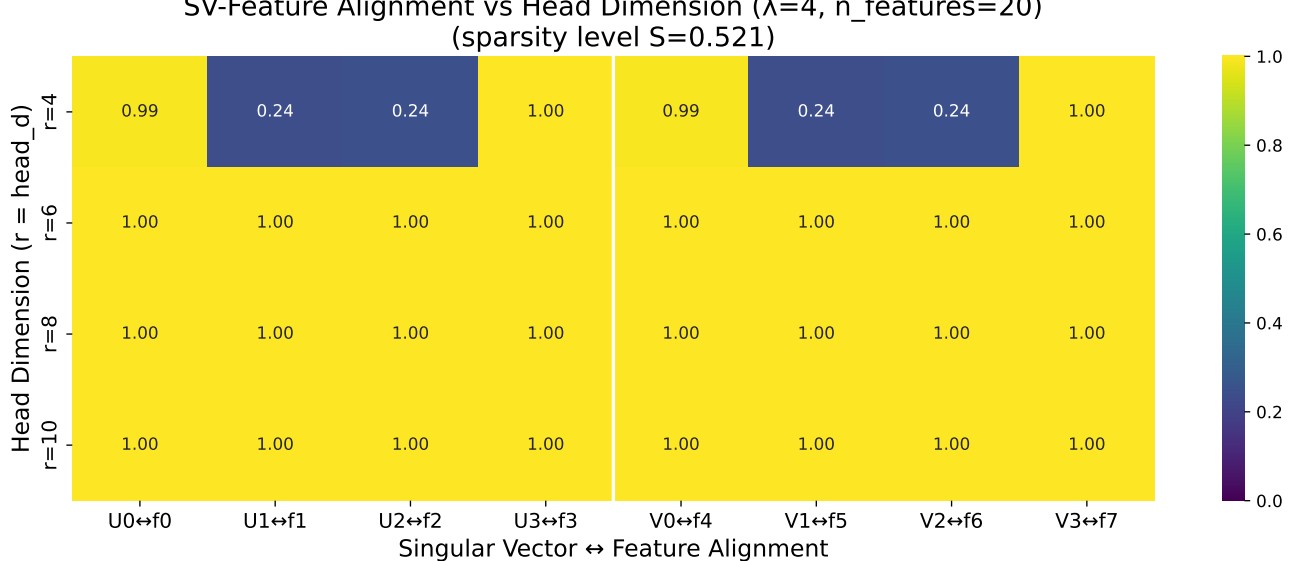

*Figure 16.* SVF Alignment is robust to varying head dimension (compared to $D = 10$).

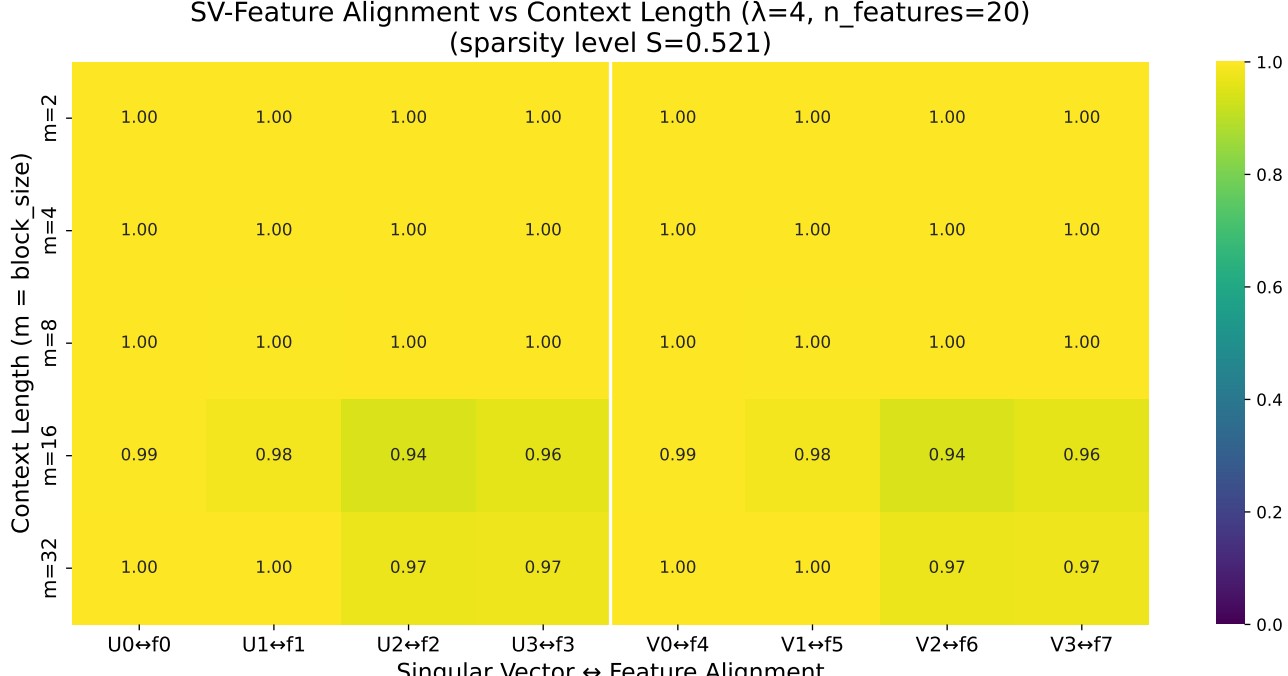

*Figure 17.* SVF Alignment is robust to varying context length.

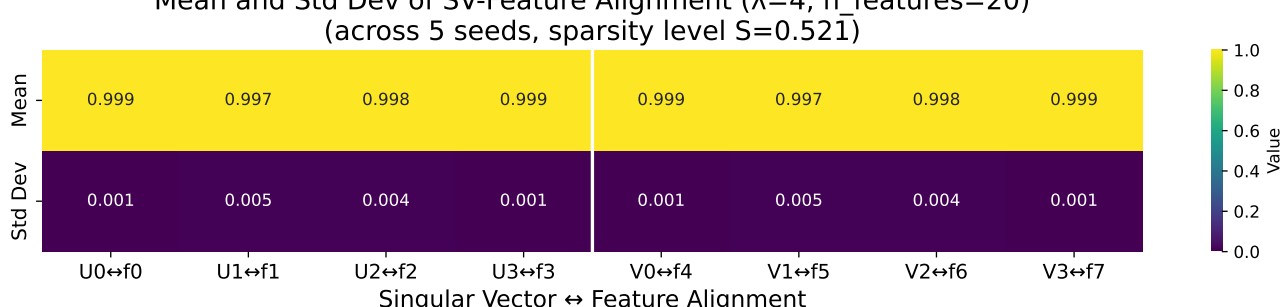

*Figure 18.* SVF Alignment is robust across a set of random seeds.

Together, these theorems provide basic theoretical justification for the phenomena observed in Section 4: alignment of features with singular vectors, and orthogonality of aligned features.

## B.2. Analysis of SVF Alignment

**Features.** In our toy model, we considered a single set of features $\{w_i\}$ which could appear in either the query or the key tokens. However, since $\Omega$ matrices are not in general symmetric, it is more general to allow the sets of features in the two tokens to differ. So in our theoretical analysis we allow the sets of features appearing in the two tokens to differ, denoting the features appearing in the query token as $\{x_i\}$ and the features appearing in the key token as $\{y_i\}$.

Hence, let $X = x_1, \ldots, x_N \subset \mathbb{R}^D$ and $Y = y_1, \ldots, y_N \subset \mathbb{R}^D$ be collections of unit vectors, which we will refer to as features.

### B.2.1. SETTING

We adopt a student-teacher setting, in which the teacher generates the target attention, and the student (model) learns via that attention pattern. We work in the context of a single head having $\Omega = W_Q^\top W_K$.

Define matrices $X = [x_1 \ \cdots \ x_N] \in \mathbb{R}^{D \times N}$, $Y = [y_1 \ \cdots \ y_N] \in \mathbb{R}^{D \times N}$, and Gram matrices

$$\Sigma_X := XX^\top \in \mathbb{R}^{D \times D}, \qquad \Sigma_Y := YY^\top \in \mathbb{R}^{D \times D}.$$

**Assumption.** In the following we assume that $\Sigma_X$ and $\Sigma_Y$ are invertible. This is expected given that we are in the regime of more features than dimensions in the hidden space. In a regime where the covariances are not invertible, $\Sigma_X^{-1}$ and $\Sigma_Y^{-1}$ can be replaced by the corresponding Moore-Penrose pseudoinverses.

**Tokens.** Fix a sampling probability $p \in (0, 1)$. Let $X_p \subseteq X$ be a random subset where each $x_i$ is included independently with probability $p$, and define the **query token**

$$r := \sum_{x \in X_p} x. \tag{3}$$

Define keys analogously: sample $m$ independent subsets $Y_p^{(1)}, \ldots, Y_p^{(m)} \subseteq Y$ (each element included i.i.d. with probability $p$) and define **key tokens**

$$s_j := \sum_{y \in Y_p^{(j)}} y, \qquad j = 1, \ldots, m. \tag{4}$$

**Student head.** Let $\Omega \in \mathbb{R}^{D \times D}$ parameterize a single attention head's bilinear score. Define student logits over keys:

$$\ell_j^{(\Omega)}(r, s_{1:m}) := r^\top \Omega s_j, \qquad j = 1, \ldots, m,$$

and student attention distribution

$$p_\Omega(j \mid r, s_{1:m}) := \frac{\exp(\ell_j^{(\Omega)})}{\sum_{k=1}^m \exp(\ell_k^{(\Omega)})}.$$

**Teacher head.** The goal of the teacher head is to output an attention value that depends on whether specific feature pairs are present. In our case, we seek to output a large attention value when features $(x_1, y_1)$ are present. These are the analogs of features $(w_0, w_1)$ in Figure 1(b).

In the body of the paper, for intuitive clarity we parameterize the teacher head using logits, defining in the case of Figure 1(b) that the teacher logit should be 1 iff features $(w_0, w_1)$ are present. We capture the setting of Figure 1(b) by defining $T \in \mathbb{R}^{N \times N}$ with $T_{11} = 1$ and $T_{ij} = 0$ everywhere else. Then the teacher head should satisfy:

$$X^\top \Omega Y = T = e_1 e_1^\top.$$

So

$$\Sigma_X \Omega \Sigma_Y = XX^\top \Omega YY^\top = XTY^\top = x_1 y_1^\top.$$

Hence we define the feature detectors
$$u := \Sigma_X^{-1} x_1, \qquad v := \Sigma_Y^{-1} y_1.$$

Given that tokens are generated as sums of features, $u^\top r$ is a (whitened) linear statistic that increases when the component $x_1$ is present in $r$ *despite correlations among features*; likewise $v^\top s_j$ increases when $y_1$ is present in $s_j$. Their product is thus a differentiable analog of "attend iff $(x_1, y_1)$ is present."

Fix a scale $\alpha > 0$ and define the teacher matrix

$$\Omega_T := \alpha u v^\top = \alpha (\Sigma_X^{-1} x_1)(\Sigma_Y^{-1} y_1)^\top.$$

Teacher logits are then:

$$\ell_j^{(T)}(r, s_{1:m}) := r^\top \Omega_T s_j = \alpha (u^\top r)(v^\top s_j),$$

and teacher attention distribution is:

$$p_T(j \mid r, s_{1:m}) := \frac{\exp(\ell_j^{(T)})}{\sum_{k=1}^m \exp(\ell_k^{(T)})}.$$

**Training objective.** The population objective is the cross-entropy of $p_\Omega$ and $p_T$:

$$\mathcal{L}(\Omega) := \mathbb{E}_{r, s_{1:m}} \left[ -\sum_{j=1}^m p_T(j \mid r, s_{1:m}) \log p_\Omega(j \mid r, s_{1:m}) \right],$$

where $(r, s_{1:m})$ are constructed via (3) and (4).

B.2.2. LEMMAS

**Lemma 1.** *Let $a, b \in \mathbb{R}^m$. Then*

$$\mathrm{softmax}(a) = \mathrm{softmax}(b) \iff a = b + c\mathbf{1} \text{ for some scalar } c \in \mathbb{R},$$

*where $\mathbf{1} \in \mathbb{R}^m$ is the all-ones vector.*

*Proof.* If $a = b + c\mathbf{1}$, then $\exp(a_i) = e^c \exp(b_i)$, and normalization cancels $e^c$. Conversely, if $\mathrm{softmax}(a) = \mathrm{softmax}(b)$, then for all $i, j$,

$$\frac{e^{a_i}}{e^{a_j}} = \frac{e^{b_i}}{e^{b_j}} \Rightarrow a_i - a_j = b_i - b_j,$$

which implies $a = b + c\mathbf{1}$. □

**Lemma 2.** *For any fixed $(r, s_{1:m})$,*

$$-\sum_j p_T(j) \log p_\Omega(j) = H(p_T) + \mathrm{KL}(p_T \| p_\Omega),$$

*so $\mathcal{L}(\Omega) = \mathbb{E}[H(p_T)] + \mathbb{E}[\mathrm{KL}(p_T \| p_\Omega)]$.*

*Proof.* Standard identity. □

**Lemma 3.** *Define differences $d_j = s_j - s_1$ for $j = 2, \ldots, m$, and let*

$$\Delta = [d_2 \ \cdots \ d_m] \in \mathbb{R}^{D \times (m-1)}, \qquad \Sigma_\Delta = \mathbb{E}[\Delta \Delta^\top].$$

*Then given that $r$ and $s_j$ are constructed via (3) and (4).*

$$\Sigma_\Delta = 2(m-1)p(1-p)\Sigma_Y.$$

*In particular, if $\Sigma_Y$ is invertible, then $\Sigma_\Delta$ is invertible.*

*Proof.* Write each key as

$$s_j = \sum_{i=1}^{N} a_{j,i} y_i,$$

with $a_{j,i} \sim \text{Bernoulli}(p)$ independent across $j, i$. Then

$$d_j = s_j - s_1 = \sum_{i=1}^{N} (a_{j,i} - a_{1,i}) y_i, \qquad \mathbb{E}[d_j] = 0.$$

For $j \geq 2$,

$$\mathbb{E}[d_j d_j^\top] = \sum_{i=1}^{N} \mathbb{E}[(a_{j,i} - a_{1,i})^2] \, y_i y_i^\top = \sum_{i=1}^{N} 2p(1-p) \, y_i y_i^\top = 2p(1-p)\Sigma_Y.$$

Now

$$\Sigma_\Delta = \mathbb{E}[\Delta\Delta^\top] = \sum_{j=2}^{m} \mathbb{E}[d_j d_j^\top].$$

There are $(m-1)$ terms, so

$$\Sigma_\Delta = 2(m-1)p(1-p)\Sigma_Y$$

So if $\Sigma_Y$ invertible and $p(1-p) > 0$ and $m \geq 2$, then $\Sigma_\Delta$ is invertible. $\qquad\square$

A similar calculation gives the query second moment

$$\Sigma_r = \mathbb{E}[rr^\top] = p(1-p)\Sigma_X + p^2 m_x m_x^\top, \quad m_x = \sum_{i=1}^{N} x_i,$$

so $\Sigma_r$ is invertible whenever $\Sigma_X$ is invertible and $p \in (0,1)$.

### B.2.3. RESULTS

**Theorem 1. Unique Minimizer is Rank-1.** *Assume $p \in (0,1)$, $m \geq 2$, $\Sigma_X$ and $\Sigma_Y$ invertible. Then the population objective $\mathcal{L}(\Omega)$ is uniquely minimized at*

$$\Omega^\star = \Omega_T = \alpha(\Sigma_X^{-1} x_1)(\Sigma_Y^{-1} y_1)^\top.$$

*Consequently, $\Omega^\star$ is rank-1 and*

$$u_1(\Omega^\star) \propto \Sigma_X^{-1} x_1, \qquad v_1(\Omega^\star) \propto \Sigma_Y^{-1} y_1.$$

*Proof.* By Lemma 2,

$$\mathcal{L}(\Omega) = \text{const} + \mathbb{E}[\text{KL}(p_T \| p_\Omega)],$$

so any minimizer must satisfy $\text{KL}(p_T \| p_\Omega) = 0$ almost surely, hence

$$p_\Omega(\cdot \mid r, s_{1:m}) = p_T(\cdot \mid r, s_{1:m}) \quad \text{a.s.}$$

Fix a sample $(r, s_{1:m})$ in this event. By Lemma 1, equality of softmax distributions implies that the corresponding logits differ by a constant shift:

$$r^\top \Omega s_j = r^\top \Omega_T s_j + c(r, s_{1:m}) \quad \forall j.$$

Subtract the equation for $j = 1$:

$$r^\top (\Omega - \Omega_T)(s_j - s_1) = 0 \quad \forall j = 2, \ldots, m.$$

In matrix form with $\Delta = [s_2 - s_1 \; \cdots \; s_m - s_1]$,

$$r^\top (\Omega - \Omega_T)\Delta = 0.$$

Multiply by $r$ on the left and by $\Delta^\top$ on the right:

$$rr^\top(\Omega - \Omega_T)\Delta\Delta^\top = 0.$$

Take expectations and use independence of $r$ and $\Delta$:

$$\mathbb{E}[rr^\top](\Omega - \Omega_T)\mathbb{E}[\Delta\Delta^\top] = \Sigma_r(\Omega - \Omega_T)\Sigma_\Delta = 0.$$

By Lemma 3, $\Sigma_\Delta$ is invertible (since $\Sigma_Y$ is invertible, $p \in (0,1)$, $m \geq 2$). Also $\Sigma_r$ is invertible (since $\Sigma_X$ is invertible and $p \in (0,1)$). Therefore,

$$\Omega - \Omega_T = 0,$$

so $\Omega^\star = \Omega_T$ uniquely.

Finally, since $\Omega_T = \alpha uv^\top$ is rank-1, its top left and right singular vectors are proportional to $u = \Sigma_X^{-1}x_1$ and $v = \Sigma_Y^{-1}y_1$. $\qquad\square$

**Corollary 1. Exact Alignment.** *Assume $p \in (0,1)$, $m \geq 2$, and the teacher is $\Omega_T = \alpha(\Sigma_X^{-1}x_1)(\Sigma_Y^{-1}y_1)^\top$ with $\alpha > 0$. Further assume that each set of features is in isotropic position (also called a* tight frame*):*

$$\Sigma_X = aI, \qquad \Sigma_Y = bI,$$

*for some positive scalars $a, b$.*

*Then the unique population minimizer satisfies*

$$\Omega^\star = \Omega_T = \frac{\alpha}{ab}x_1 y_1^\top.$$

*In particular, $\Omega^\star$ is rank-1 and its unique nonzero left and right singular vectors satisfy*

$$u_1(\Omega^\star) \parallel x_1, \qquad v_1(\Omega^\star) \parallel y_1.$$

*Proof.* By Theorem 1 $\Omega^\star = \Omega_T$. Under the assumption of isotropy $\Sigma_X^{-1} = \frac{1}{a}I$ and $\Sigma_Y^{-1} = \frac{1}{b}I$, hence

$$\Omega^\star = \alpha(\Sigma_X^{-1}x_1)(\Sigma_Y^{-1}y_1)^\top = \alpha\left(\frac{1}{a}x_1\right)\left(\frac{1}{b}y_1\right)^\top = \frac{\alpha}{ab}x_1 y_1^\top.$$

Thus $\Omega^\star$ has left and right singular vectors proportional to $x_1$ and $y_1$. $\qquad\square$

**Theorem 2. Approximate Alignment.** *Let*

$$\Sigma_X = XX^\top = \frac{N}{D}(I + E_X), \qquad \Sigma_Y = YY^\top = \frac{N}{D}(I + E_Y), \tag{5}$$

*for some symmetric "error" matrices $E_X, E_Y \in \mathbb{R}^{D \times D}$. Assume the sets are close to isotropic with $\max\{\|E_X\|_2, \|E_Y\|_2\} < 1/2$. Define $\tau = 4\|E_X\|_2\|E_Y\|_2 + 2\|E_X\|_2 + 2\|E_Y\|_2$. Then if $\tau < 1$,*

$$\sin\angle(u_1, x_1), \ \sin\angle(v_1, y_1) \ < \ \frac{\tau}{1 - \tau},$$

*and in particular if $\tau < 1/2$,*

$$\sin\angle(u_1, x_1), \ \sin\angle(v_1, y_1) \ < \ 8\|E_X\|_2\|E_Y\|_2 + 4\|E_X\|_2 + 4\|E_Y\|_2. \tag{6}$$

We note that the operator norm bound on $E_X$ also establishes a bound on feature interference, because $XX^\top$ and $X^\top X$ have the same spectra.

*Proof.* From Theorem 1 the minimizer is

$$\Omega^\star = \alpha \Sigma_X^{-1} x_1 y_1^\top \Sigma_Y^{-1}.$$

From (5), we have

$$\Sigma_X^{-1} = \frac{D}{N}(I + E_X)^{-1}, \qquad \Sigma_Y^{-1} = \frac{D}{N}(I + E_Y)^{-1}.$$

Therefore

$$\Omega^\star = \alpha \left(\frac{D}{N}\right)^2 (I + E_X)^{-1} x_1 y_1^\top (I + E_Y)^{-1}. \tag{7}$$

We will rewrite $\Omega^\star$ as

$$\Omega^\star = \lambda (x_1 y_1^\top + \Delta), \qquad \lambda := \alpha \left(\frac{D}{N}\right)^2, \tag{8}$$

where $\Delta$ collects all terms that prevent $\Omega^\star$ from being exactly rank 1 in the direction $x_1 y_1^\top$. Then

$$\Delta = (I + E_X)^{-1} x_1 y_1^\top (I + E_Y)^{-1} - x_1 y_1^\top \tag{9}$$

Defining $A = (I + E_X)^{-1}$ and $C = (I + E_Y)^{-1}$, (9) can be rewritten as

$$\Delta = (A - I) x_1 y_1^\top (C - I) + (A - I) x_1 y_1^\top + x_1 y_1^\top (C - I) \tag{10}$$

To bound $\Delta$ in operator norm, an important first step is to bound $\|A - I\|_2$ and $\|C - I\|_2$. Since by assumption $\|E_X\|_2 < 1$ and $\|E_Y\|_2 < 1$, the inverses can be expanded in a Neumann series

$$(I + E_X)^{-1} = I - E_X + E_X^2 - \dots, \qquad (I + E_Y)^{-1} = I - E_Y + E_Y^2 - \dots,$$

which converges absolutely in operator norm. So we can bound $\|A - I\|_2$:

$$\|A - I\|_2 = \|\sum_{k=1}^\infty (-E_X)^k\|_2 \leq \sum_{k=1}^\infty \|E_X\|_2^k = \frac{\|E_X\|_2}{1 - \|E_X\|_2}$$

and similarly for $\|C - I\|_2$. Since we assume $\|E_X\|_2, \|E_Y\|_2 \leq 1/2$:

$$\|A - I\|_2 \leq \frac{\|E_X\|_2}{1 - \|E_X\|_2} \leq 2\|E_X\|_2. \tag{11}$$

Next, to bound the influence of $\Delta$, we use submultiplicativity of the operator norm applied to (10):

$$\|\Delta\|_2 \leq \|A - I\|_2 \|x_1 y_1^\top\|_2 \|C - I\|_2 + \|A - I\|_2 \|x_1 y_1^\top\|_2 + \|x_1 y_1^\top\|_2 \|C - I\|_2$$

and noting (11) as well as that $\|x_1 y_1^\top\|_2 = 1$:

$$\|\Delta\|_2 \leq 4\|E_X\|_2 \|E_Y\|_2 + 2\|E_X\|_2 + 2\|E_Y\|_2. \tag{12}$$

To bound the deviation of the singular vectors of $\Omega^\star$ from $x_1, y_1$, we use a standard singular-vector perturbation theorem. Recalling (8):

$$\Omega^\star = \lambda (x_1 y_1^\top + \Delta),$$

let $u_1, v_1$ be the top left and right singular vectors of $\Omega^\star$. Applied to our setting, Wedin's theorem (P.-Å.Wedin, 1972) states that

$$\sin \angle(u_1, x_1) \leq \frac{\|\Delta\|_2}{\delta} \tag{13}$$

where $\delta = \sigma_1(x_1 y_1^\top) - \sigma_2(x_1 y_1^\top + \Delta)$. Weyl's inequality Horn & Johnson (2012, Theorem 4.3.1) shows that $|\sigma_2(x_1 y_1^\top) - \sigma_2(x_1 y_1^\top + \Delta)| \leq \|\Delta\|_2$, so $\delta \geq 1 - \|\Delta\|_2$. So as long as $\|\Delta\|_2 < 1$,

$$\sin \angle(u_1, x_1), \; \sin \angle(v_1, y_1) \; < \; \frac{\|\Delta\|_2}{1 - \|\Delta\|_2}$$

and in particular if $\|\Delta\|_2 \leq 1/2$, then

$$\sin \angle(u_1, x_1), \; \sin \angle(v_1, y_1) \; < \; 8\|E_X\|_2 \|E_Y\|_2 + 4\|E_X\|_2 + 4\|E_Y\|_2.$$

$\square$

## B.3. Feature Orthogonalization

Next we show that when features vary and $\Omega$ is fixed, interference between features will result in orthogonalization of features.

### B.3.1. SETTING

For clarity we analyze the setting with two $X$ features $(x_1, x_2)$ and two $Y$ features $(y_1, y_2)$. The query token $r$ can be either $x_1$ or $x_2$, and the key token can be either $y_1$ or $y_2$. We assume that SVF alignment has taken place for features $(x_1, y_1)$, meaning that the top singular vectors of $\Omega$ are $(u_1, v_1) = (x_1, y_1)$. We will show that under these conditions, if $(x_2, y_2)$ are allowed to vary, they will become orthogonal to $(u_1, v_1)$.

Since there are only two possible key tokens, softmax reduces to a sigmoid on a logit gap $\delta := \ell_1 - \ell_2$. In the two-token case, $p_\Omega(1) = \sigma(\delta)$ with $\sigma(t) = \frac{1}{1+e^{-t}}$. The teacher sets a probability $p^\star$ and training uses two-key cross-entropy:

$$\mathrm{CE}(p^\star, \sigma(\delta)) = -p^\star \log \sigma(\delta) - (1 - p^\star) \log(1 - \sigma(\delta)). \tag{14}$$

We assume the teacher matrix is rank-2:

$$\Omega_T = \sigma_1 u_1 v_1^\top + \sigma_2 u_2 v_2^\top, \qquad \sigma_1 > \sigma_2 > 0, \tag{15}$$

with orthonormal $u_1, u_2$ and $v_1, v_2$.

Note that $(x_1, y_1) = (u_1, v_1)$. Let $x_2, y_2$ be trainable unit vectors. We consider two contexts, each with two key tokens $s_1 = y_1$ and $s_2 = y_2$. In each context, we define the *teacher gap* $\delta^\star$ which is the difference of logits between $r^\top \Omega y_1$ and $r^\top \Omega y_2$. This is the parameterization of the attention head.

We focus on establishing the orthogonality of $y_2$ and $y_1$.

To capture the need for accurate reconstruction of inputs, we penalize interference between features. Thus our overall training objective is

$$\mathcal{J}(x_2, y_2) = \mathrm{CE}(\delta_A^\star, \delta_A(x_2, y_2)) + \mathrm{CE}(\delta_B^\star(x_2, y_2)) + \frac{\lambda}{2}(y_2^\top y_1)^2 \tag{16}$$

where the student gaps are

$$\delta_A(x_2, y_2) = x_1^\top \Omega_T(y_1 - y_2), \qquad \delta_B(x_2, y_2) = x_2^\top \Omega_T(y_1 - y_2)$$

### B.3.2. RESULTS

**Theorem 3. CE minimization with reconstruction loss forces orthogonalization.** *Let $(x_2^\lambda, y_2^\lambda)$ be any global minimizer of* (16)*, with the constraints $\|x_2\| = \|y_2\| = 1$. Then*

$$y_2^{\lambda\top} y_1 \leq \sqrt{\frac{2}{\lambda}\left(\mathcal{J}_\lambda(x_2', y_2') - \inf_{x_2, y_2}[\mathrm{CE}(\delta_A^\star, \delta_A(x_2, y_2)) + \mathrm{CE}(\delta_B^\star, \delta_B(x_2, y_2))]\right)}$$

*for any comparison pair $(x_2', y_2')$ with $\|x_2'\| = \|y_2'\| = 1$. In particular, if there exists a feasible $(x_2', y_2')$ achieving minimal CE while also having $y_2'^\top y_1 = 0$, then the optimizer must satisfy $y_2^{\lambda\top} y_1 = 0$*

*Proof.* Fix any $\lambda > 0$. For any $(x_2, y_2)$ we can decompose the objective as

$$\mathcal{J}(x_2, y_2) = \underbrace{\mathrm{CE}(\delta_A^\star, \delta_A(x_2, y_2)) + \mathrm{CE}(\delta_B^\star, \delta_B(x_2, y_2))}_{=: \mathcal{L}_{\mathrm{CE}}(x_2, y_2)} + \frac{\lambda}{2}(y_2^\top y_1)^2$$

Let $(x_2^\lambda, y_2^\lambda)$ be a global minimizer.

Now choose any comparison pair $(x_2', y_2')$ with $y_2'^\top y_1 = 0$. Optimality gives

$$\mathcal{L}_{\mathrm{CE}}(x_2^\lambda, y_2^\lambda) + \frac{\lambda}{2}(y_2^{\lambda\top} y_1)^2 \leq \mathcal{L}_{\mathrm{CE}}(x_2', y_2') + \frac{\lambda}{2}\underbrace{(y_2'^\top y_1)^2}_{=0}.$$

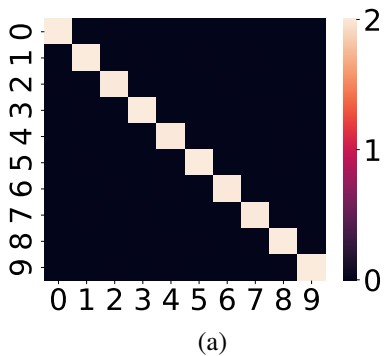
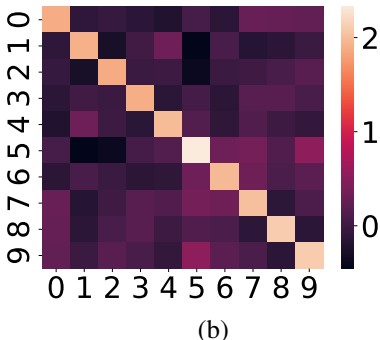

(a)                      (b)

*Figure 19.* $WW^\top$ matrices showing isotropic arrangement of features (a) without attention, and (b) when two features are of interest to the head.

Rearranging yields the finite-$\lambda$ bound

$$\frac{\lambda}{2}(y_2^{\lambda\top}y_1)^2 \le \mathcal{L}_{\text{CE}}(x_2', y_2') - \mathcal{L}_{\text{CE}}(x_2^\lambda, y_2^\lambda) \le \mathcal{L}_{\text{CE}}(x_2', y_2') - \inf_{x_2, y_2} \mathcal{L}_{\text{CE}}(x_2, y_2)$$

which establishes the result. □

## C. Isotropy

### C.1. Isotropy in Toy Model

Here we show evidence of the isotropy of features in the toy model. We consider two cases, corresponding to Figures 1(a) and (b). In Figure 19(a) we show $WW^\top$ for the case where no attention head is present. The figure shows that the features are arranged isotropically. In Figure 19(b) we show $WW^\top$ for the case where two features are of interest. Here, features are still arranged approximately isotropically, suggesting that the requirements of Corollary 1 and Theorem 2 for subsequent SVF alignment can still be met over the remaining features.

### C.2. Anisotropy of SAEs in GPT-2

To assess realistic degrees of feature anisotropy in a real model, we examine a GPT-2 sparse autoencoder (SAE) dictionary.

For each of GPT-2 Small's 12 layers, we load the residual-stream SAEs from (Bloom, 2024). These consist of $N = 24{,}576$ dictionary elements in $D = 768$ dimensions. We row-normalize the decoder $W_{\text{dec}}$, and compute the feature Gram matrix $\Sigma_X = W_{\text{dec}}^\top W_{\text{dec}}$ together with the deviation-from-isotropy operator $E_X = (D/N)\Sigma_X - I$, which is the same operator used to characterize anisotropy in Theorem 2.

The results in Figure 20 show that GPT-2 features are anisotropic, consistent with prior work (Ethayarajh, 2019; Gao et al., 2019; Li et al., 2025) with values of $\|\Sigma_X\|_2$ ranging from 10 to 55.

### C.3. Alignment under Anisotropy

**Setup.** We test Theorems 1 and 2 in a standard configuration of our toy model ($N = 20$ features, $D = 10$ hidden dimensions and head dimension $H = 10$). In this experiment, we freeze features and test whether singular vectors align with them. The feature dictionary $W \in \mathbb{R}^{N \times D}$ is constructed so that in the absence of anisotropy, singular vectors can align exactly. Specifically, $W^\top W = (N/D)I$ exactly: eight features of interest occupy their own orthogonal subspace at norm $\sqrt{2}$, and twelve auxiliary features form six antipodal pairs in the remaining two-dimensional subspace at norm $\sqrt{1/3}$. A random unit vector $u \in \mathbb{R}^D$ is drawn and the dictionary is distorted by

$$W_c = W\big(I + (\sqrt{1+c} - 1)\,uu^\top\big),$$

yielding $\Sigma_X = W_c^\top W_c = (N/D)(I + c\,uu^\top)$, so $\|E_X\|_2 = c$ by construction. We sweep $c$ over values in $[0, 50]$ reflecting the range of values of $\|E_X\|_2$ seen in GPT-2 SAEs (previous subsection).

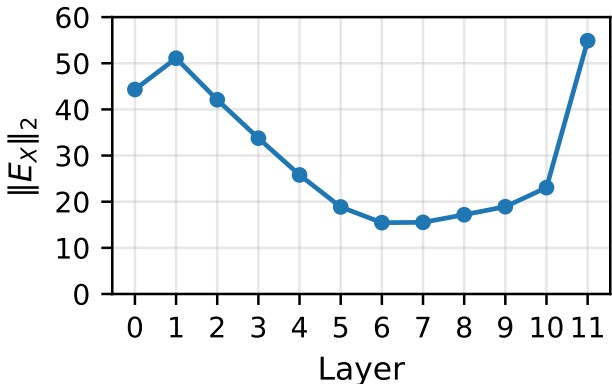

Figure 20. Measured values of $\|E_X\|_2$ for SAEs trained on each layer of GPT-2 range between 10 and 55.

**Training.** For each $c$, the dictionary is frozen and the head matrices $W_Q, W_K \in \mathbb{R}^{H \times D}$ are trained for 20,000 steps with AdamW at a constant learning rate of $10^{-3}$ on a pure softmax-matching loss (no reconstruction term). The loss assigns four feature pairs $(\ell, r) \in \{(0,4), (1,5), (2,6), (3,7)\}$ using standard target attention logits $\{24, 21, 18, 15\}$. We use the standard attention loss for batches of one query and $K = 4$ candidate keys (i.e., cross-entropy between the softmax of these target logits and the softmax of the scaled-dot-product attention scores). The procedure is repeated for five random $u$.

**Measurement.** After training we form $\Omega = W_Q^\top W_K = U\Sigma V^\top$ and, for each pair, measure the maximum cosine alignment of $w_\ell$ against the columns of $U$ and of $w_r$ against the columns of $V$, averaged across the four pairs. Figure 4 reports mean $\pm$ one standard deviation across runs. The figure plots three quantities against $c$: the raw alignment $\max_j |\langle \hat{w}, u_j \rangle|$, the anti-whitened alignment $\max_j |\langle \hat{w}, \Sigma_X u_j / \|\Sigma_X u_j\| \rangle|$, and the Theorem 3 lower bound $\sqrt{1 - s^2}$ where $s = \tau/(1 - \tau)$ and $\tau = 4c^2 + 4c$ (shown only where $\tau < 1$).

## D. SVF Alignment Can Occur Under RoPE

The analysis in Theorems 1 – 3 assumes a model in which a fixed QK matrix $\Omega$ is used to compute attention on tokens $r, s$ regardless of the positions of those tokens in the context window. However RoPE (Su et al., 2024) has become the de facto positional encoding in modern LLMs (Dubey et al., 2024; Jiang et al., 2023; Team et al., 2024; Bai et al., 2023). Here we present a preliminary study showing how SVF alignment occurs in a model that uses RoPE.

RoPE extends the QK matrix by including a position-dependent rotation of each token. For query and key tokens $r, s \in \mathbb{R}^D$ at positions $p_r, p_s \in \{1, \ldots, m\}$ the attention logit under RoPE is

$$\ell(r, s; p_r, p_s) = \left\langle R_{p_r} W_Q r, \ R_{p_s} W_K s \right\rangle, \tag{17}$$

where $R_p$ is RoPE's block-diagonal rotation (acting on the $H$-dimensional head vector) by angles $\omega_j p$ on the $j$-th coordinate pair, with frequencies

$$\omega_j = \mathrm{base}^{-2j/H}, \qquad j = 0, 1, \ldots, H/2 - 1.$$

For a token pair at positions $p_r, p_s$ we have $\Omega^{(p_r - p_s)} = W_Q^\top R_{p_r}^\top R_{p_s} W_K$. Thus the $\Omega$ used in Theorems 1 – 3 corresponds, under RoPE, to $\Omega^{(0)}$.

Note that we can think of the vector $q = W_Q r \in \mathbb{R}^H$ as a concatenation of $H/2$ two-dimensional pieces, $q = (q^{(0)}, q^{(1)}, \ldots, q^{(H/2-1)})$ with each $q^{(j)} \in \mathbb{R}^2$. $R_p$ acts on each piece independently as the 2D rotation $R(\omega_j p)$, i.e. $(R_p q)^{(j)} = R(\omega_j p) q^{(j)}$. So the logit (17) decomposes band-by-band:

$$\ell(r, s; p_r, p_s) = \sum_{j=0}^{H/2-1} \left\langle q^{(j)}, R(\omega_j(p_r - p_s)) k^{(j)} \right\rangle, \tag{18}$$

with $q = W_Q W f^{(r)}$ and $k = W_K W f^{(s)}$.

**Experimental Setup.** We use $m = 16$, base $= 100$, giving five bands:

$$(\omega_0, \ldots, \omega_4) \approx (1.000, \ 0.398, \ 0.158, \ 0.063, \ 0.025).$$

The corresponding periods $2\pi/\omega_j$ are $(6.28, 15.8, 39.7, 100, 251)$; relative to $m = 16$, band 1's period essentially equals the context length.

In our experiment we study both position-independent and position-dependent features.

The target logit for a query/key pair at $(p_r, p_s)$ combines terms for position-independent features and zero or more *positional pair* terms:

$$\ell^T(r, s; p_r, p_s) \ = \ \sum_{(i,j) \in \mathcal{F}} T_{ij} \, f_i^{(r)} f_j^{(s)} \ + \ \sum_\kappa A_\kappa \, \cos(\omega_\kappa(p_r - p_s)) \, f_{r_\kappa}^{(r)} f_{s_\kappa}^{(s)}. \tag{19}$$

$\mathcal{F}$ is the set of position-independent feature indices – for these, we use three pairs $(0, 3)$, $(1, 4)$, $(2, 5)$ with logit strengths $T_{ij} = 24, 18, 15$. We use one positional pair, so $\kappa = \{1\}$. The positional feature pair $(r_1, s_1) = (6, 7)$; we match it to band 1, $\omega_1 = 0.398$, with logit strength $A_1 = 21$.

**Results.** Figure 21 shows alignment of singular vectors and features under RoPE. Note that attention is computed using the logits defined by (17), but the singular vectors used here are those of $\Omega = \Omega^{(0)}$. Nonetheless we observe that, for features that are not position-dependent, singular vector alignment is strong (just as in the non-RoPE case). Additionally, we note that the positional feature pair is *also* aligned with singular vectors.

We can gain further insight into how the model jointly arranges features and weights by noting the following. For a position-$p$ RoPE rotation $R_p$, let $R_p^{(j)}$ be its rows $2j, 2j + 1$ for $j = 0, 1, \ldots, H/2 - 1$. Then in computing logits (Equation 17) $R_p^{(j)}$ acts only the head's encoding vectors $W_Q^{(j)}, W_K^{(j)}$ (rows $2j, 2j + 1$ of the Q and K matrices). Thus the magnitude of a feature's projection into the 2D subspaces $W_Q^{(j)}, W_K^{(j)}$ determines the extent to which the head is positionally-sensitive to the feature with frequency $\omega_j$. For small $j$, the head is relatively sensitive to positional changes to the feature, and for large $j$, the head is relatively insensitive to positional changes to the feature.

Figure 22 shows the projection of features into each subspace of $W_Q^{(j)}$ and of $W_K^{(j)}$. The figure shows the model has aligned features $w_i$ with subspaces $W_Q^{(j)}, W_K^{(j)}$ according to positional sensitivity. Position-insensitive features $w_0, \ldots, w_5$ are allocated to subspaces with low positional sensitivity (low-frequencies $\omega_2, \ldots, \omega_4$). On the other hand, positionally-sensitive features $w_6$ and $w_7$ are allocated to frequency $\omega_1$.

Intuitively, this happens because allocating features $w_6$ and $w_7$ to any band $j \neq 1$ would produce a term in (19) oscillating at frequency $\omega_j$. Since the training data samples $p_r, p_s$ uniformly from $1, \ldots, m$, the mismatch between $\cos(\omega_j \Delta p)$ and $\cos(\omega_1 \Delta p)$ prevents an exact match between $\ell$ and $\ell^T$. Hence gradient descent will drive the feature to the band 1 subspace. On the other hand, a position-independent feature has no $\Delta p$ dependence. No RoPE band has $\omega_j = 0$ exactly, so such features cannot be represented exactly through a single band. However for bands with small $\omega_j$ (high index $j$, slow rotation) the error over the context range $1, \ldots, m$ is small. Thus the model allocates position-independent features to high-$j$ bands.

# E. More Features of Interest than Head Capacity

The experiments in Section 4 show that the toy model assigns one feature pair to each singular vector direction. However, when the number of feature pairs is greater than $H$, this is no longer possible because the rank of $\Omega$ is $H$. Here we present a preliminary study of the nature of SVF alignment when there are more features of interest to the head than can be represented orthogonally in $\mathbb{R}^H$.

To study this setting we constrain the representational capacity of the head by setting $H = 5$. We then study the case in which the number of features of interest ranges from $H$ (5) to $2H$ (10). As usual, the target logits decline linearly with the pair index. Here, $\ell^T$ of pair $i$ is set to $1 +$ the number of features $- i$.

In Figure 23(a), we show alignment of the five singular vectors of the head with the features for each case. Surprisingly, the model assigns all of the features having the lowest target logits to the smallest singular vector (singular vector with the smallest singular value). Figure 23(b) shows that this effect persists when model capacities are higher ($H = 10$, features of interest range from 10 to 20).

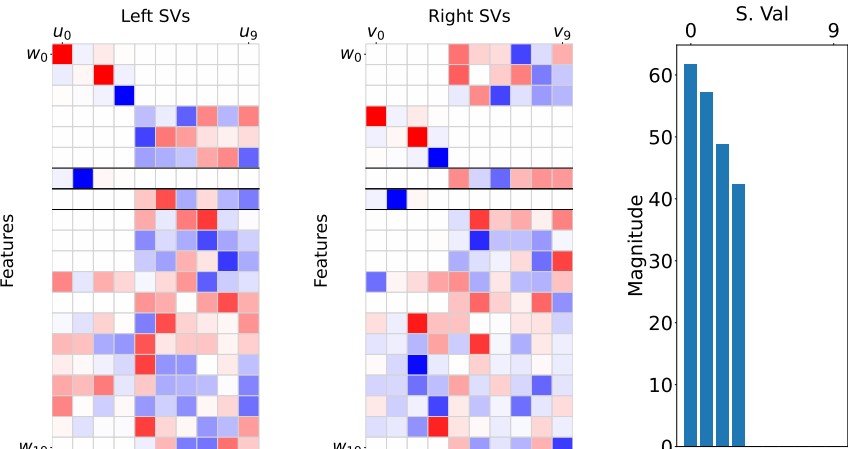

*Figure 21.* Singular vectors of $\Omega^{(0)}$ align with features under RoPE. Cosine similarities of singular vectors and features, and magnitudes of singular values. Feature pair (6, 7) (marked with black lines) has varying target logit depending on position of tokens; all other feature pairs have position-independent target logits.

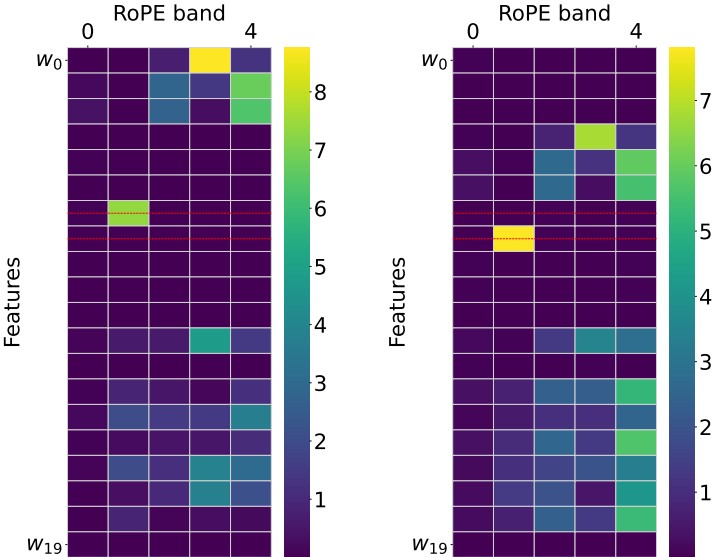

*Figure 22.* Features align with the head's encodings according to RoPE band. Magnitude of projection of each feature $w_i$ into the query-side and key-side 2D subspaces of band $j$ ($W_Q^{(j)}$ and $W_K^{(j)}$). Dashed red lines denote positional features 6 and 7.

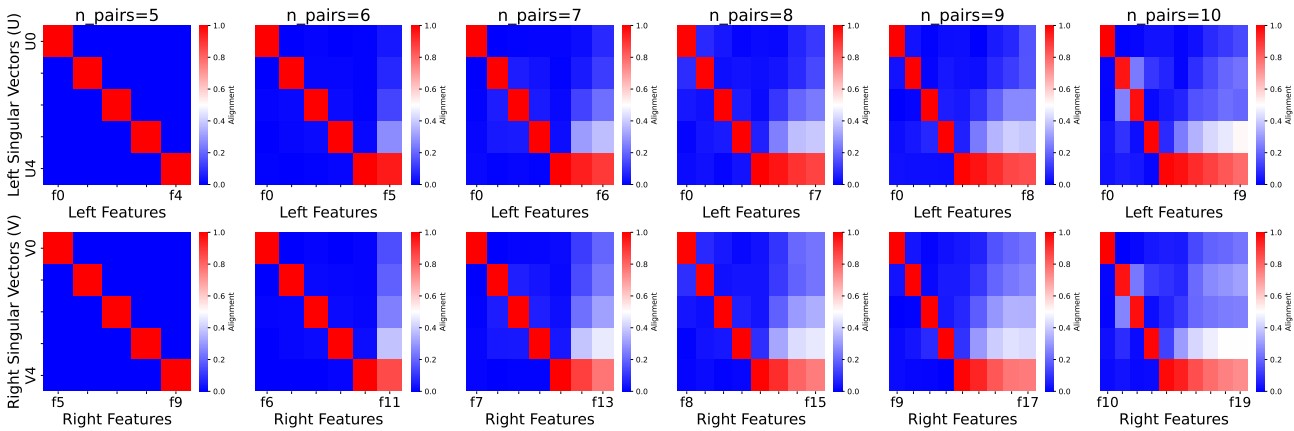

(a) $N = 20$ features, hidden dimension $D = 10$, head dimension $H = 5$.
Number of feature pairs of interest varies from 5 (head capacity) to 10 (model capacity).

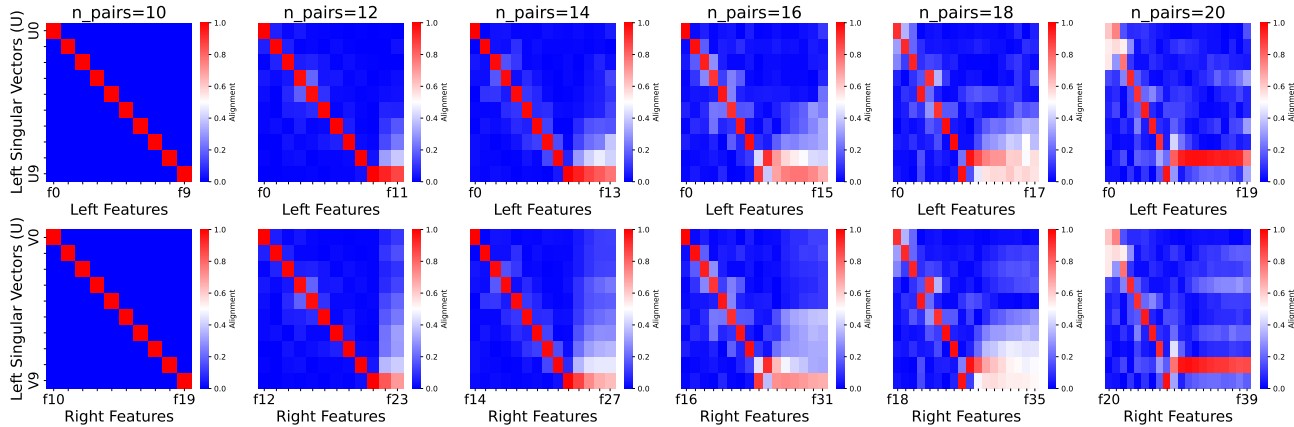

(b) $N = 40$ features, hidden dimension $D = 20$, head dimension $H = 10$.
Number of feature pairs of interest varies from 10 (head capacity) to 20 (model capacity).

*Figure 23.* When there are more features of interest than head capacity, features are superposed primarily on the smallest singular vector, and most singular vectors still map to a single feature. Absolute cosine similarity of features and singular vectors. We show absolute value of cosine similarity for clarity due to sign ambiguity of SVD.

This behavior has a number of consequences. It means that the model cannot distinguish the features that mapped to the same singular vector, and in fact can mistakenly associate features from different pairs. On the other hand, the model still allocates high-logit features one-to-one with singular vectors. As a result, most singular vectors of the model still correspond to features. We note that sparse decomposition will still arise for a model in this setting, although less robustly.

Clearly, more study is needed to understand the behavior of SVF alignment when the number of features of interest to the head exceeds its capacity, but these results give an initial indication that SVF alignment still can have utility in this regime.

## F. Sparse Attention Decomposition in Toy Model Indicates that Features of Interest are Present

Here we show that in the toy model, sparse attention decomposition occurs only when at least one feature pair of interest is present. To show this, we consider the complementary case to Figure 5. Figure 24 shows that when features of interest are *not* present, sparsity of attention decomposition is *absent*.

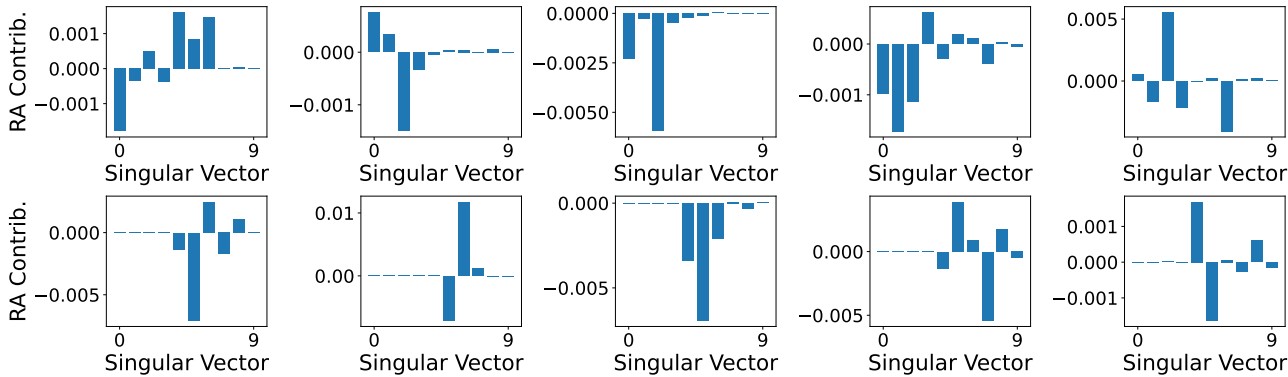

*Figure 24.* When features of interest are not present, attention is not sparsely decomposable. Top: Token pairs without features of interest, early in training; Lower: Late in training. Compare to Figure 5.

## G. Pythia Experiments

### G.1. Details of Experiments

Here we provide details of our study of sparse attention decomposition in Pythia (Section 5). We used the following prompt from the Indirect Object Identifiation task (Tigges et al., 2024): *Then, Simon and Andrew were working at the restaurant. Simon decided to give a basketball to.* The highest-logit output token is "Andrew", indicating that the model successfully completes the task.

We denote the tokens that participate in the circuit as follows: 'Simon': S1; 'and': S1+1; 'Andrew': IO; 'Simon' (second occurrence): S2; and 'to' (second occurrence): END. We use the heads and token pairs identified as participating in the circuit by (Tigges et al., 2024). Table 2 lists the heads and token pairs used.

Our results in Figures 7(b) and 9(b) are averaged over the heads in Table 2, and smoothed with a window size of 8.

| Name | Layer | Head | Destination Token | Source Token |
|---|---|---|---|---|
| Previous Token | 2 | 6 | S1+1 | S1 |
| Induction 1 | 4 | 11 | S2 | S1+1 |
| Induction 2 | 4 | 6 | S2 | S1+1 |
| Induction 3 | 5 | 0 | S2 | S1+1 |
| Name Mover 1 | 10 | 7 | END | IO |
| Name Mover 2 | 9 | 4 | END | IO |
| Name Mover 3 | 8 | 2 | END | IO |
| Name Mover 4 | 8 | 10 | END | IO |
| S-Inhibition 1 | 7 | 9 | END | S2 |
| S-Inhibition 2 | 6 | 6 | END | S2 |
| S-Inhibition 3 | 7 | 2 | END | S2 |
| Positive Copy Suppression 1 | 8 | 9 | END | S2 |

*Table 2.* Attention heads used in studying Pythia, Section 5

### G.2. Relative Attention Sparsifies as a Result of Training

In Figure 25 we present relative attention decompositions for all of the heads studied in Pythia, at the start and end of training. These heads and token pairs were identified as participating in the IOI circuit in (Tigges et al., 2024).

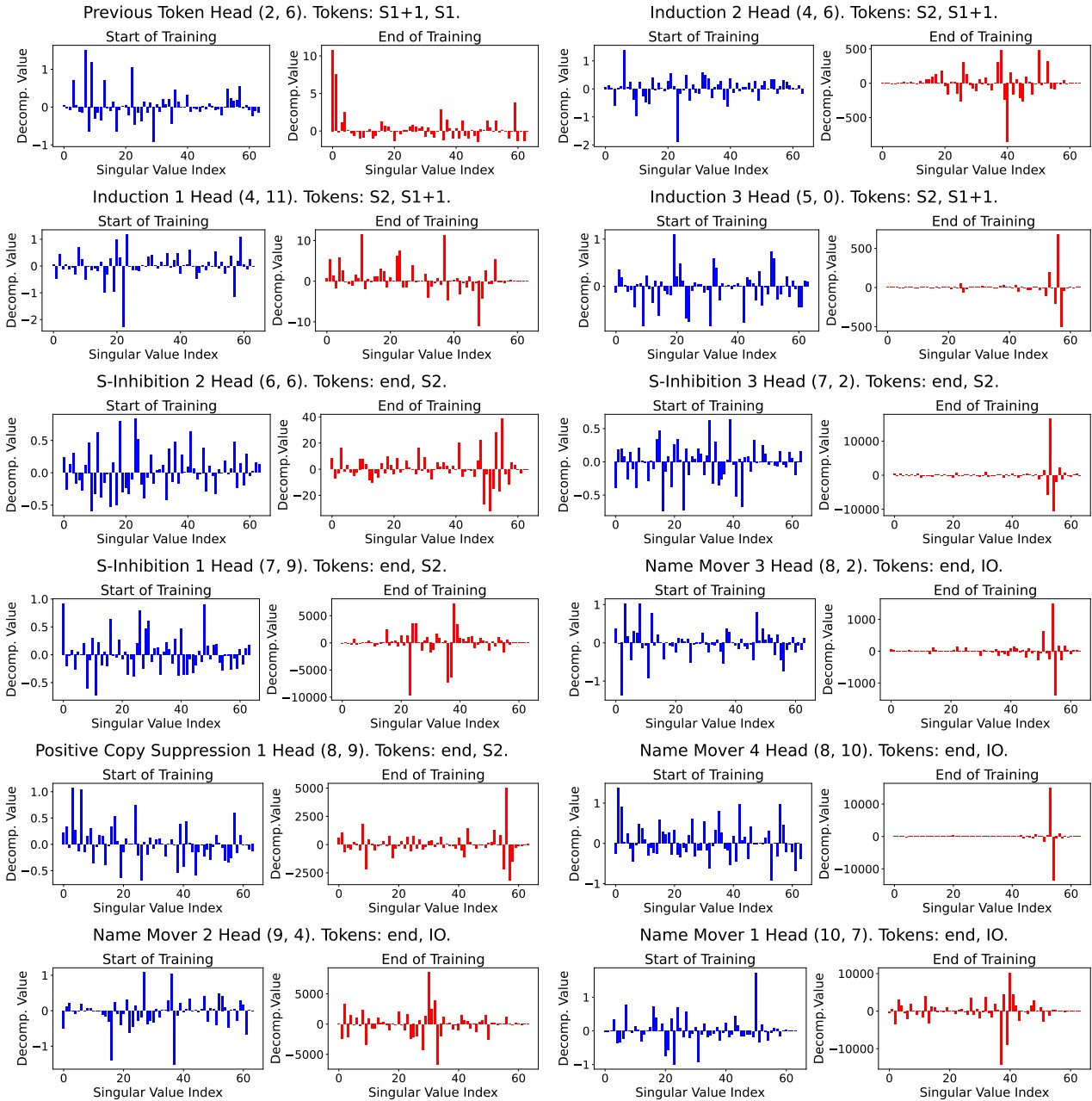

*Figure 25.* Sparse decomposition in Pythia generally increases during training. Each plot shows decomposition of relative attention at the start and end of training.

