# OpenReview forum: "Singular Vectors of Attention Heads Align with Features"
_ICML.cc/2026/Conference — ICML 2026 regular_

### Official Review · Reviewer_PcAD · 2026-03-10

**Soundness:** 4
**Presentation:** 4
**Significance:** 3
**Originality:** 3
**Overall Recommendation:** 5
**Confidence:** 4

**Summary:**

This paper studies the question of alignment (high cosine similarity) between the singular vectors of the “attention matrix” (product of key/value matrices, since these are always used together) and “feature vectors”, roughly the vectors that all inputs into the model are weighted sums of these vectors. While this phenomenon has been observed previously / studied to some extent, this paper is the systematically / rigorously demonstrate scenarios when it occurs. They take as the main starting pointing a previously studied model of Elhage et al 2022, a model which learns feature vectors and the weights of a single layer RELU network at the same time, with them being related— the features form a matrix W and the weights of the forward network are tr(W); in this setting, with learning meaning reconstructing the weights / composition of the input over the feature vectors, they demonstrate that the features converge to an orthogonal “isometry” of the space. This paper adds “attention” to this model: in addition to learning features / its transpose = a hidden feedforward layer and trying to reconstruct the composition of the input, there is also an “attention loss” where the attention head needs to output (via softmax) attention scores as close as possible to target scores, parametrized by a matrix T which determines how much the attention should be increased for each pair of features in the query and key. The total loss is a weighted sum of the two losses (reconstructing the input’s decomposition over the features, and the new/added attention loss). They show both experimentally, and theoretically in several theorems, that in this model one obtains convergence (alignment / high cosine similarity) between feature vectors and singular vectors of the attention head. A particularly interesting experiment within these shows that in the case that T is such that multiple pairs of features matter in the target attention, singular vectors for the highest weighted pairs evolve first, and so on in order of the weights given in T. In the last section, the authors point out that a prediction of this theory is that for tokens with high attention scores, the logits would would be a sparse weighted sum over the singular values of the attention matrix; they test this in several small attention / language models and find this to be the case consistently, with such sparsity evolving over training.

**Compliance With Llm Reviewing Policy:**

Affirmed.

**Key Questions For Authors:**

Nice paper, a few additional questions / points that I wonder if you have considered / one could add somewhere (perhaps in discussion)
1) this one is probably the easiest / good to clarify in the paper - what changes if you do not have reconstruction, you just have attention head(s)? It seems like that reconstruction loss in part drives orthogonal feature vectors, and then attention SVD bends towards them. Since most attention models have only that, clarifying this point would make the paper clearer / more relevant.

2) how do your results interact with multiple layers — it seems like they are mostly for a 1 layer (or just 1 head) scenario (I think it’s fine to just have understood 1 layer for now— but thinking about how this might be extended is important since most models use multiple layers)



MINOR (not "important questions"):
Critical, e.g., to build -> in order to build

Worth mentioning some intuition; since attention is indeed linear operation on the inputs

Use non bracket versions of cite when its’ inline (\citer)

Since previous work finds, for example, that inter-layer communication can be understood as taking places in singular value subspaces, and other related work, there is better phrasing than saying that this is an “assumption” in previous work. Perhaps previous work found some experimental evidence suggesting this, but no work has done a detailed investigation and theoretical investigation to determine when this happens

Line 160 - it’s easy to misread it as “f = the model”, I propose the following rewrite: “An input f to the model is …”. Also there is something unclear in this paragraph — “a universe of N features w_i” implies that they are fixed (as does “W is the matrix whose columns are the features w_i”, but then it says W is learned… if I understand correctly from your description, there are no predetermined features in Elhage et al 2022, just the weights W. In this case I’d remove saying “a universe of N features”, since the model / encoder itself is determining the “tokens” (or rephrase similarly)

A bit confused at 181 (beginning of section 4)- how is this experiment different from Elhage 2022, is it identical (it’s fine if it is, as a reproduction / sanity check, but should be clearer). Also, as I understood a linear layer + RELU, using W^T as the weights for that layer, is *also* included in the model with attention, that’s what is meant by L_reconstruction. It should made a bit clearer that your new model is two models in parallel - the previous one, where W^T is recycled as the linear / hidden layer, and next to it an attention head, with the loss being a weighted sum of the two

207/208 - I suggest “the first left singular vector” and “the first right singular vector” rather than 0-indexing

I know it may be a matter of space but I would have liked (shortened) statements of Theorems 2 / 3 in the main text (in boxes for example), since they seem elegant and nice and worth highlighting! Btw why is the first one indexed at 2?

“Analyzing logits” — while I get the general picture, there is some imprecision about how the term “sparsity” is used… a single real number l(res) cannot be sparse or not… perhaps all large logins can be encoded sparsely over the SVD basis, or something like this, more precision would make this clearer

Line 419 - briefly add, 8(a) are distributions over what? Over different tokens from the test set?

**Limitations:**

Yes

**Strengths And Weaknesses:**

Overall this paper is a nice contribution in the theory of mechanistic interpretability of attention models. It says that if inputs can be understood as (roughly) linear combinations of features, attention heads will train to have an SVD aligned with these feature vectors.
Soundness - paper is very robust / sound. This includes theoretical results, and experiments which are varied and substantiate the claims clearly.

Presentation - well written, clear, very good overall. I’ve included a detailed list of nits / suggestions / small improvements that can make the paper even better (under Questions -- note that I put the main / significant questions first, then the small nits, as there wasn't a separate place to put them)

Significance - a nice contribution to the very important topic of interpretability. However a minor weakness is that the results aren’t necessarily so surprising; one might expect attention matrices to compute some sort of dot products with features if those determine the needed output (attention scores), this is especially the case when features are learned at the same time with the same features being the hidden layer computing a reconstruction of the input. It is also specifically relevant to one layer (it is less clear how this generalizes beyond that), and assumes inputs are linear combos of features. So in that sense it not earth-shattering, but I see it as a necessary grounding of an important principle, with which we can go further / extend to more cases.
Originality - good, thoroughly establishes a basic principle of alignment between attention head matrices and representations

---

> ### Author Rebuttal · Authors · 2026-03-30
>
> Thank you for your comments, which are helpful.
>
> ## Question 1
>
> In an attempt to answer this question, we ran the following experiment. We run the toy model in the default setting, but disabling the reconstruction loss.  Interestingly, we find that the model still aligns singular vectors with features, and features still orthogonalize.  That suggests that attention loss is sufficient to drive SVF alignment on its own, at least in certain settings.  What we find however is that a greater range of target logits is needed for the SVF alignment to converge compared to the default setting.   This confirms that the reconstruction loss normally helps the features maintain distinct directions, which in turn gives the attention head enough structure to align SVs even with closely spaced target logits.  Without that anchor, the logits themselves need to carry the distinguishing signal, and this requires a greater range among the target logits. This result is a helpful addition to the paper which we will include in the revision.  (And in our response to reviewer YZx3 we show that when starting from isotropic features with reconstruction loss disabled, SVF alignment happens exactly).
>
> ## Question 2
>
> SVF alignment would interact across layers through its effect on feature representations.  If a given feature were important for multiple heads across different layers, then SVF alignment could cause singular vectors of different layers to themselves be aligned.  We note also that [Merullo et al, 2024] shows that singular vectors of OV matrices in earlier layers can be aligned with singular vectors of QK matrices in later layers. We believe that this is an excellent direction for future work.
>
> ## Minor questions
>
> We thank the reviewer for the helpful suggestions which will improve the style and readability of the paper. We will incorporate all these changes in the final manuscript, and we will answer the ones that we believe that need specific answer.
>
> > Since previous work finds, for example, that inter-layer communication can be understood as taking places in singular value subspaces, and other related work, there is better phrasing than saying that this is an “assumption” in previous work. Perhaps previous work found some experimental evidence suggesting this, but no work has done a detailed investigation and theoretical investigation to determine when this happens
>
> We agree (this point was also raised by YZx3).  We will remove the use of “assumed” in the paper and point out that prior work demonstrated empirical evidence for SVF alignment.
>
> > 207/208 - I suggest “the first left singular vector” and “the first right singular vector” rather than 0-indexing
>
> Indeed this would be clearer - we will do so in the revision.
>
> > I know it may be a matter of space but I would have liked (shortened) statements of Theorems 2 / 3 in the main text (in boxes for example), since they seem elegant and nice and worth highlighting! Btw why is the first one indexed at 2?
>
> Thank you for the suggestion – we agree and will do so.  (This point was raised by other referees.)
>
> > “Analyzing logits” — while I get the general picture, there is some imprecision about how the term “sparsity” is used… a single real number l(res) cannot be sparse or not… perhaps all large logins can be encoded sparsely over the SVD basis, or something like this, more precision would make this clearer
>
> We will clarify this language in the revision.
>
> > Line 419 - briefly add, 8(a) are distributions over what? Over different tokens from the test set?
>
> This point was also raised by multiple referees.  Due to an oversight on our part, we neglected to describe the GPT-2 experiments in sufficient detail.  The GPT-2 results in Figure 8(a) are averaged over 128 prompts that vary names, templates, and objects.  We will clarify that in the revision.

---

> > ### Author Rebuttal · Reviewer_PcAD · 2026-04-03
> >
> > Thank you for your detailed response and the extra experiments. Adding the new experiment you ran to the paper will make it stronger and more relevant! I will maintain my score.

---

### Official Review · Reviewer_VqPv · 2026-03-12

**Soundness:** 4
**Presentation:** 2
**Significance:** 3
**Originality:** 1
**Overall Recommendation:** 3
**Confidence:** 2

**Summary:**

This paper explores the question of how the singular vectors of the product of query and key weight matrices align with input features, using a carefully designed experimental setup. Based on the empirical findings, the paper shows that these singular vectors gradually evolve during training to become aligned with the input features. This observation may provide useful insights into the interpretability of attention-based language models.

**Compliance With Llm Reviewing Policy:**

Affirmed.

**Key Questions For Authors:**

I have included the main questions for the authors in the “Strengths and Weaknesses” sections above.

**Limitations:**

yes

**Strengths And Weaknesses:**

Strengths:
1.	The paper investigates an interesting question: how the query-key weight matrices align with input features during training.
2.	A deeper understanding of attention matrices could potentially benefit a wide range of Transformer-based models, which have become the de facto standard in many modern machine learning applications.
3.	The paper presents its arguments in a clear, step-by-step manner with toy example and real language model.
4.	The empirical analysis appears to provide evidence supporting the authors’ main claims.

Major Weaknesses:
1.	Heavy reliance on empirical observations:
Although the paper claims to explore the phenomenon from a theoretical perspective, much of the argument appears to rely on empirical observations. It would be helpful if the authors could provide a clearer theoretical statement that explains the observed alignment behavior in a unified way.
In this sense, it remains unclear what fundamentally new insight this work provides compared to prior studies. The paper itself notes that previous work has already observed that “features used by an attention head tend to be aligned with its singular vectors” (3rd paragraph in the Introduction). Given this, the authors should clarify what new theoretical or empirical contribution this work makes beyond confirming this phenomenon.
2.	Marginal novelty on top of previous works’ observation:
A.	Strongly related prior work such as [1] shows that deep linear networks trained with gradient descent recover the dominant singular modes of the input-output correlation matrix. This implies that the learned mapping (stacked weight matrices $W_nW_{n-1}…W_1$) naturally aligns with task-relevant feature directions present in the data distribution.
In the toy setup of the current paper, the inputs are generated as random combinations of underlying features that appear to form an isotropic basis. Under such conditions, it seems plausible that the learned query-key matrices would align with these feature directions even without any special mechanism.
Therefore, it would be helpful if the authors clarify to what extent this paper’s observed feature alignment goes beyond what would already be expected from classical deep linear network learning dynamics. This paper considers a bilinear form (as in the attention mechanism), but I believe this setting is not fundamentally different from the linear network case.
B.	Furthermore, the previous work [2] empirically demonstrated that attention tends to become sparse during training. The discussion in Section 5 (“Sparse Attention Decomposition”) appears to revisit a similar idea. In addition, the experimental setup (e.g., model architecture and dataset) seems largely identical to that used in [2].
Given this similarity, it would be helpful if the authors could clarify how the contributions of this section differ from those of the previous work, and more explicitly articulate the novelty of the proposed analysis.
3.	Potentially advantageous experimental settings
A.	In the toy experimental setup, the model is optimized not only with a reconstruction loss but also with an additional attention loss. This attention loss directly encourages the model to follow a specific attention pattern, which does not appear to reflect a realistic training setting. It is possible that this direct supervision contributes to the clear emergence of the SVF phenomenon shown in Figure 2. I wonder how much this additional loss influences the observed alignment behavior on top of the reconstruction loss only.
B.	In Appendix F, the authors appear to select specific tokens in the prompt (e.g., “Simon”, “and”, “Andrew”, etc.) for the analysis. My understanding is that the reported relative attention scores are primarily computed with respect to these selected tokens. I would appreciate clarification on how these tokens were chosen and whether the results generalize beyond these specific examples.
Minor Weaknesses / Suggestion:
1.	It would be helpful to include an illustrative diagram of the toy example model architecture to improve readability for the readers.
2.	Description related to Phytia-based IOI task experiment (in 5. Sparse Attention Decomposition) needs more details, such as specific attention layer of measuring relative attention, and so on. I suggest authors to show illustrative image that explains the whole pipeline of this analysis.
The description of the Pythia-based IOI task experiment (Section 5, Sparse Attention Decomposition) would benefit from additional details. For example, it would be useful to mathematically explain how the authors computed the relative attention scores given a set of hidden states. An illustrative figure explaining the overall analysis pipeline would also improve clarity.
3.	In the fourth paragraph of ‘3.Methods’ section, index notation $j$ was used to denote token positions (1~$m$). However, in the following paragraph it appears to refer to the index of features. Using different notation for these two quantities would improve clarity.
References:
[1] Saxe, Andrew M., James L. McClelland, and Surya Ganguli. "Exact solutions to the nonlinear dynamics of learning in deep linear neural networks." arXiv preprint arXiv:1312.6120 (2013).
[2] Franco, Gabriel, and Mark Crovella. "Sparse attention decomposition applied to circuit tracing." arXiv preprint arXiv:2410.00340 (2024).

---

> ### Author Rebuttal · Authors · 2026-03-30
>
> We appreciate the comments on the paper, which will improve it.  We respond below.
>
> ## Weakness 1
>
> Prior work has indeed observed SVF alignment, and we do not claim to have discovered the phenomenon itself. Our contribution is to provide a theoretical foundation for this empirically observed phenomenon and a controlled environment for studying its mechanics.  As we write, “no previous work has presented a detailed investigation of why and when SVF alignment occurs.”  In support of that, Theorems 1 through 4 characterize conditions under which SVD alignment provably occurs.   This theoretical foundation is important to form a reliable basis for future work.  We will revise the introduction to make this more explicit, and we will add more detailed summaries of the theorems in the main body of the paper to clarify their contribution.
>
> ## Weakness 2A
>
> We agree that there can be a connection between what is known about deep linear networks and our results. However we don’t think that the results from [1] directly predict our results, because the model we are considering is quite far from a classical linear network. We are considering the attention mechanism: a nonlinear function (softmax) applied to the outputs of a set of bilinear (not linear) forms sharing a common left-side input. There is no obvious mapping from our setup to a linear network of the kind analyzed in [1]. Thus we believe that our work adds a new dimension to the understanding of the role of singular modes in deep networks.
>
> In addition, as we explain in the introduction, previous papers have empirically observed SVF alignment without suggesting any proposed mechanism, so it does not appear that previous authors have concluded that the alignment of singular vectors with features would be an obvious consequence of the attention mechanism. We appreciate being able to connect with the literature on linear networks, and will add a citation to [1] in our background and discuss the differences between the attention mechanism and a linear network.
>
> ## Weakness 2B
>
> Indeed we do not introduce the idea of sparse attention decomposition, as we make clear on lines 082-083, and lines 288-289.   We adapt SAD to our setting in order to make three contributions that go beyond [2]. First, we argue that if SVF alignment holds, then SAD should hold. Second, we use SAD to explore the training dynamics of Pythia and compare them with the training dynamics of the toy model.  This adds supporting evidence to our conclusions. Third, the experiments in Section 5 allow us to rule out a potential alternative explanation for attention decomposition sparsity. Specifically, we have found that the most common alternative explanation that comes to mind is that it could be caused by the concentration of the QK matrix spectrum. By rotating the singular vectors without changing the QK matrix spectrum, we address this concern.
>
> ## Weakness 3A
>
> We appreciate the opportunity to clarify the experimental setup.  Attention loss is computed for each pair of tokens presented to the head.  Loss is determined based on whether the two tokens presented to the head contain the specific feature pair, which occurs at random. The supervision is deliberate, because the question we study is how a head, that must compute attention based on feature pairs, allocates its singular vectors.  As we say in the paragraph headed “Why this model?” the setting we adopt here reflects a minimal, commonly-accepted view of model internals and the Linear Representation Hypothesis (LRH).
> Regarding whether the additional loss influences the observed alignment behavior, we show in Figure 1(a)  the arrangement of features in the absence of attention loss.  In this case there is no SVF alignment present (these results are not shown – we will add them in the revision).
>
> ## Weakness 3B
>
> Due to an oversight on our part, we neglected to describe the GPT-2 experiments in sufficient detail. The GPT-2 results in Figure 8(a) are averaged over 128 prompts that vary names, templates, and objects. We will clarify that in the revision.
>
> ## Minor Weaknesses / Suggestion 1
>
> We agree (we actually had such a figure but removed it for space).  We will add it to the revision.  The figure is here: [https://anonymous.4open.science/r/rebuttal-svf-alignment-icml-6C73/toy_model_improved.pdf](https://anonymous.4open.science/r/rebuttal-svf-alignment-icml-6C73/toy_model_improved.pdf)
>
> ## Minor Weaknesses / Suggestion 2
>
> We agree.  Although we present some of these details in Appendix F, we understand that the specifics of how we compute relative attention and SAD for the Pythia experiment would benefit from greater detail.  We will also add a figure explaining the overall analysis pipeline.
>
> ## Minor Weaknesses / Suggestion 3
>
> Thank you for pointing this out; we will use different notations for the two quantities in the revision.

---

> > ### Author Rebuttal · Reviewer_VqPv · 2026-04-06
> >
> > I appreciate the authors' point-by-point response. While the clarifications were helpful, two critical issues remain unaddressed: the lack of a rigorous theoretical foundation and the limited novelty of the proposed approach.

---

> > > ### Author Response · Authors · 2026-04-06
> > >
> > > Thank you for the follow-up. We would like to address the two concerns highlighted in your acknowledgement: the lack of a rigorous theoretical foundation and the limited novelty of the proposed approach.
> > >
> > > Rigorous theoretical foundation: our paper’s contribution is precisely to **provide a formal account of *why* and *when* SVF alignment occurs**. The submission does not rely on empirical evidence alone: **it contains four explicit theorems that characterize the mechanism behind the phenomenon**. Theorems 1–4 show that singular vectors recover whitened features in general, raw features exactly under isotropy, approximately under bounded anisotropy, and that reconstruction pressure induces feature orthogonalization. **These results are the paper’s core theoretical contribution. They go beyond confirming an already observed empirical effect by identifying concrete conditions under which SVF alignment should be expected.**
> > >
> > > Novelty: We study the attention mechanism, **a highly nonlinear computation that is not analogous to a linear network. Prior work has not proposed an explanation for SVF alignment**, and our proposed explanation **does not arise** from the same mechanism as in linear networks. Our novelty is therefore not merely observing alignment, but explaining why and when it arises in attention heads.
> > >
> > > For these reasons, **we believe the current submission addresses the two concerns raised in the acknowledgement**, and we agree that the revised paper should make these points more explicit in the main text.

---

### Official Review · Reviewer_YZx3 · 2026-03-13

**Soundness:** 2
**Presentation:** 2
**Significance:** 3
**Originality:** 3
**Overall Recommendation:** 3
**Confidence:** 4

**Summary:**

The paper studies singular vector-feature (SVF) alignment, i.e., the cosine similarity between the singular vectors of an attention head's QK matrix and the linear feature representations that the head attends to, specifically why and when it should hold.

**Compliance With Llm Reviewing Policy:**

Affirmed.

**Final Justification:**

Through the rebuttal I increased my original score from 2 to 3 as I think the paper can benefit from clearer presentation of its theory and core arguments used to explain the experiments in the paper. The rebuttal by the authors has addressed some of my concerns, and I do think the paper needs to be revised in terms of the presentation of its core arguments about SAD and singular value (not rank) control, which is clearer through the chronology of experiments the authors describe in their second rebuttal reply.

**Key Questions For Authors:**

- Is it possible to test the prediction of the bound in Theorem 3? For eg, can it be shown (say through future work) that alignment quality in real models correlates with the anisotropy of the representation space in the toy setting? Would it be possible to discuss these implications in current work?
- If I understand correctly, there is a single prompt being used for the Pythia/GPT-2 experiments, right? How sensitive are the SAD results to the specific prompt? Are there any results that average over multiple IOI prompts with different names, or if this doesn't make sense to perform, can you explain the choice?

**Limitations:**

Yes

**Strengths And Weaknesses:**

*Strengths*

(Significance)
 - Though it has been shown earlier that inter-layer communication occurs through low-rank subspaces defined by singular vectors, the formal conditions under which SVF alignment is guaranteed, and the controlled toy-model environment that makes alignment directly observable is novel and adds motivation for SVF alignment.
- Theorem 4 and corresponding experiments show features of interest to a head are driven to become orthogonal to each other to minimise CE with reconstruction, while attention heads drives them to align with singular vectors.
- The finding under superposition when features of interest exceed head capacity that the least important features collapse onto the smallest singular vector whereas more important ones can align one-on is interesting and could be emphasised more in the main text along with its implications.


*Weaknesses*

(Soundness, Presentation, and Originality)
- I am not sure what insights the sparse attention decomposition (SAD) results in real models (Pythia, GPT-2 on IOI) are adding in comparison to existing work (Merullo et al, 2024) who (from what I understand) already showed that SVD-derived subspaces are causally important in the same models on the same task with a stronger result under interventions. Can you explain the relevance of the SAD experiments in comparison and add this to the section on real model experiments? Also how does SAD connect to feature identification?
- I think the core contribution of the paper could be reframed to emphasise that it provides a theoretical foundation for an empirically observed phenomenon (which Merullo et al. provided empirical evidence for in real models and not just "assumed"), plus a controlled environment for studying its mechanics.
- For a paper thus with its main contributions being a bit theoretical, it would help emphasise the implications of each theorem and the intuitions behind the proofs even briefly in the main text, so the novelty of the contributions is clearer. Right now, it's not very evident how the theorems connect with the experiments (based on my current understanding).
- The theoretical results are based on either fixing the features and optimising the QK matrix $\Omega$ or the way around, but even in the toy experiments, both change together. Then how do the toy experiments validate the theory, or how do the theoretical insights translate to the same training dynamics?
- Also, in real models, features would be roughly anisotropic, right? If not, please clarify in the text why the assumption of feature isotropy holds for real model experiments. How anisotropic are real model activations in the layers where you measure SAD? If Theorem 3's bound is loose or the anisotropy is large, the theoretical guarantee may not meaningfully apply. And what are the "cone directions", can they resolve this tension? I could not find a connection being made to it anywhere, but it was just briefly mentioned.
- The scalability argument that SVF-based feature identification is cheaper than SAEs is compelling in principle, but there are no experiments that test whether the phenomenon holds at scale. If not within the scope of current work to conduct these experiments, how can you justify this argument?
- What does it mean that a rotation matrix control rules out low-rank properties? Isn't it full rank? It's rather that rotation would destroy the directional alignment of singular vectors with token features while preserving the spectrum exactly. So the experiment distinguishes directional alignment from a few large spectral values dominating, which means the claim would be that sparsity arises from specific directions of the singular vectors, not from the spectral distribution of the QK matrix. This is not a rank issue. The claim should be reframed in terms of directional alignment versus spectral concentration.

(Merullo et al, 2024: https://arxiv.org/abs/2406.09519)

I am happy to adjust the score if these issues and questions can be addressed convincingly.

---

> ### Author Rebuttal · Authors · 2026-03-30
>
> We thank the referee for the careful read and very helpful comments.
>
> * [SAD] Indeed, [Merullo et al, 2024] showed that SVD-derived subspaces are causally important in GPT-2 and Pythia applied to IOI (and other tasks) and we will make that clear in the revision of Sections 2 & 5.
>
>     We include the SAD experiments to make three points: they add supporting evidence to our conclusions by comparing the dynamics of Pythia during training with the dynamics of the toy model (Figure 6); they allow us to show visually how striking sparsity in attention decomposition is in a real model (Figure 7 top and mid); and they allow us to rule out a potential alternative explanation for attention decomposition sparsity (concentration of QK matrix spectrum),  Figure 7 lower.
>
> * [Core Contrib] We agree and will revise the contributions to read as suggested here.  We also agree that Merullo et al. empirically demonstrated alignment, and we will remove reference to “assuming” the phenomenon.
> * [Emphasize Thms] We agree. The discussion of the next two points makes clear that we need to better connect the theorems to the paper body (eg, as discussed under the next two bullet points).
>
> * [Interplay of Thms and Dynamics] Figure 3 shows what happens in the toy experiments.
>
>     The heatmap for feature 0 shows that feature 0 does not change much over the course of training;  the heatmap for left SV 0 shows that SV 0 moves into final position around time 1500. The alignment plot (lower) shows that the SV 0’s movement brings it into alignment with feature 0. This is what Thms 1 and 2 predict: SV 0 shifts to align with feature 0.
>
>     Next, we see SV 1 shifting over the period up to time 4000, at which point feature 1 abruptly shifts. This is what Thm 4 predicts: to minimize reconstruction loss, feature 1 shifts to a position orthogonal to feature 0. The process then repeats for the remaining features and singular vectors.
>
>     We will expand this discussion and add it to the revision.
>
>
> * [Anisotropy] We agree that in real models, features are likely to be anisotropic. As a rough estimate, we measured anisotropy using the dictionary elements of an SAE as a proxy for model features (Bloom SAEs on GPT-2 Small).   We measure anisotropy using $\Vert E_X \Vert_2$ which is the control variable in Theorem 3. [This figure](https://anonymous.4open.science/r/rebuttal-svf-alignment-icml-6C73/sae_E_norm.pdf) shows that $\Vert E_X \Vert_2$ ranges from 15 to 55.
>
>     However, we emphasize that the presence of anisotropy does **not** necessarily destroy SVF alignment. In terms of the paper, we are using isotropy as a simplifying assumption to show how Theorem 1 applies to the toy model.
>
>     Thm 1 says that, **regardless of anisotropy of features,** SVs recover whitened features, ie, $u = \Sigma_X^{-1} x.$  Thm 2 then says that if features are perfectly isotropic, raw features are recovered exactly. We then argue that, at least in the toy model, minimizing reconstruction error tends to lead to isotropic features. In real models, where features are anisotropic, the question is open how closely SVs align with raw features. There are challenges in answering this question: access to true features, determining which features are of interest to a head, and determining whether the features of interest to the head are isotropic – excellent questions for future study.
>
>     Thm 3 shows that small deviations from isotropy lead to small recovery errors; however, it is less useful as a predictive tool as it is a loose bound.
>
>     To illustrate, we run an experiment based on a standard config of the toy model. We start with isotropic features, freeze the features, and train the head weights.  We then introduce controlled anisotropy into the feature set, varying $c = \Vert E_X\Vert_2$ over the range of values found in GPT-2 above. [This figure](https://anonymous.4open.science/r/rebuttal-svf-alignment-icml-6C73/anisotropy_alignment.pdf) shows that even when anisotropy gets quite large (to max seen in GPT-2), SV recovery is quite good (cos sim only down to about 0.75 for $c = 50$).
>
>     The figure also shows how loose the Thm 3 bound is, and it demonstrates Thm 1: even when features are anisotropic, recovery is quite good for whitened singular vectors.
>
>     Happy to provide omitted details in followup. We will expand these points in the revised manuscript.
>
> * [Scalability] SVF-based feature identification is used in [Franco & Crovella, 2025] to trace circuits and seems to achieve accuracy similar to other methods; [Pan et al., 2025] showcase feature identification via SVs across many different vision transformers.
>
> * [Rank] We are using low-rank to mean low effective rank – that is, some singular values are quite small compared to the largest; we will clarify in the revision.
>
> * [Prompt] We neglected to describe the GPT-2 experiments in detail.  The GPT-2 results in Figure 8(a) are averaged over 128 prompts that vary names, name order, and objects.

---

> > ### Author Rebuttal · Reviewer_YZx3 · 2026-04-03
> >
> > Thank you for clarifying the role and dynamics of anisotropy, and for clarifying the scope of the contributions. However, I still do not get the rank issue mentioned in the paper and tested through rotation matrix control. It would be really helpful if you could address that in the followup. I will increase the score I gave earlier and am happy to revisit it again.

---

> > > ### Author Response · Authors · 2026-04-04
> > >
> > > We appreciate the reviewer’s attention to our responses and the fact that they increased their score as a result.  We’re happy to explain more regarding the rank issue and the basis-rotation experiment.
> > >
> > >
> > > We included this experiment in the paper because of early feedback we got on the sparse attention decomposition (SAD) experiments.  To recap, SAD means that a few terms in the sum (1) [line 302] are large compared to the others. Each term in equation (1) includes one singular value, but it also depends on the alignment of the tokens with the corresponding singular vectors. Hence there was concern that SAD could be caused simply by a few large singular values, rather than by alignment between singular vectors and features.
> > >
> > > To mitigate this concern we use the rotation experiment.  In this experiment, we apply a random  rotation matrix to the singular vectors $U$ and $V$ and then recompute the terms in (1).  This changes the alignment of singular vectors with features, without changing the magnitudes of the singular values. The fact that SAD disappears under this random rotation mitigates the concern about SAD arising solely due to magnitudes of singular values. This shows that the sparsity is tied to the specific directions of the singular vectors, as would be expected under SVF alignment.
> > >
> > > We agree that using the term “low rank” to describe the presence of a few large singular values was inaccurate and confusing here and we will correct this in the revision.  For example, we will correct lines 375-376 (right side) to read:
> > > “Sparse attention decomposition is not attributable simply to the presence of a few large singular values in $\Omega$,” and we will remove references to “low rank” throughout.

---

### Official Review · Reviewer_4oH1 · 2026-03-13

**Soundness:** 3
**Presentation:** 3
**Significance:** 3
**Originality:** 3
**Overall Recommendation:** 5
**Confidence:** 3

**Summary:**

This paper investigates why and when the singular vectors of attention heads align with model features in language models. Prior interpretability work often assumes that features used by attention heads correspond to directions in the singular vector decomposition (SVD) of the attention matrix, but this assumption had not been theoretically justified.

The authors show both theoretically and empirically that such singular vector–feature (SVF) alignment naturally arises during training. The paper also introduces Sparse Attention Decomposition (SAD), a prediction of SVF alignment stating that relative attention can be decomposed into a small number of singular vector components.

**Compliance With Llm Reviewing Policy:**

Affirmed.

**Key Questions For Authors:**

1. How broadly do you think these results generalize beyond the GPT-2/Pythia and IOI-style settings studied in the paper?
2. Do you think the conclusions here would still hold when analyzing groups of interacting heads instead of one head at a time?

**Limitations:**

yes

**Strengths And Weaknesses:**

Strentghs:
1. The paper combines toy models, theoretical analysis, and experiments on language models (Pythia), which provides convincing and well-rounded evidence for the proposed SVF alignment hypothesis.
2. The paper studies how singular vectors and features evolve during training, showing that alignment emerges gradually, which gives deeper insight into the learning process of attention heads.
3. The paper is easy to follow, with well-designed figures and visualizations that help illustrate feature geometry, singular vector alignment, and sparsity dynamics.

Weakness:
1.The rotation experiments do suggest that the sparsity is not happening just because of low-rank structure, but I still feel the paper does not fully rule out other possible reasons behind this behavior. For example, things like feature anisotropy, architectural bias, or even the specific task/circuit they study might also be affecting the decomposition patterns.

---

> ### Author Rebuttal · Authors · 2026-03-30
>
> Thank you for reading our paper closely and providing these comments.
>
> ## Weakness 1
>
> We agree that these are valid points. We acknowledge that we have not ruled out anisotropy as a factor in sparse attention decomposition, and we mention this in the Limitations section. We discuss the impacts of anisotropy in detail in our response to reviewer YZx3. There, we make two points. First, Theorem 1 shows that even in the presence of anisotropy, features can be recovered from covariance-preconditioned singular vectors. Second, the presence of anisotropy does not necessarily destroy alignment. We show that even in the presence of anisotropy of comparable magnitude to what is found in GPT-2, the toy model is able to recover singular vectors with reasonable accuracy (cosine similarity of 0.75;  see our response to YZx3 for details). We propose to include results along these lines in the revision.
>
> We do attempt to mitigate architectural bias somewhat by illustrating the same effect in both Pythia and GPT-2. For example, with respect to the attention mechanism, Pythia uses RoPE, as well as parallel attention and MLPs; while GPT-2 uses global positional embeddings, with sequential attention and MLPs.  However, it would be beneficial to study sparse attention decomposition in more models, and we will mention that in the revision.
>
> With respect to the task studied, we acknowledge that we did not study a range of tasks in the paper. We note that previous work has shown evidence of SVF alignment in the Laundry List task [Merullo et al, 2024], the Greater-Than and Gender-Pronoun tasks [Ahmad et al, 2025, Franco & Crovella, 2025], and in vision tasks [Pan et al, 2025].
>
> ## Question 1
>
> There is evidence that SVF alignment holds for other tasks. As noted above, there is evidence of SVF alignment in the Laundry List task [Merullo et al, 2024], the Greater-Than and Gender-Pronoun tasks [Ahmad et al, 2025, Franco & Crovella, 2025], and in vision tasks [Pan et al, 2025]. The work in [Franco & Crovella, 2025] additionally shows evidence in Gemma-2, and the work in [Pan et al, 2025] shows evidence of this phenomenon in multiple vision transformers (ViTs). Specifically, they study 16 different ViT models from 6 families, with different training objectives.
>
> ## Question 2
>
> We believe that if multiple heads make attention decisions that are independent functions of feature presence, then SVF alignment should occur as shown in the paper. However, in the case where attention heads make coordinated or correlated attention decisions based on feature presence, it is not clear whether SVF alignment would occur as cleanly as it occurs in the paper. The prior work that we cite [Merullo et al, 2024, Franco & Crovella, 2025, Pan et al, 2025, Ahmad et al, 2025] suggests that SVF alignment does occur often in real settings. However, the question is a fascinating direction for further work, one that could be approached both via simulation and theory, following the framework we use in the paper. We will add a short discussion of this direction as potential future work.
>
> [Ahmad et al, 2025]: Ahmad, Areeb, Abhinav Joshi, and Ashutosh Modi. "Beyond Components: Singular Vector-Based Interpretability of Transformer Circuits." The Thirty-ninth Annual Conference on Neural Information Processing Systems.
>
> [Franco & Crovella, 2025]: Franco, Gabriel, and Mark Crovella. "Pinpointing attention-causal communication in language models." The Thirty-ninth Annual Conference on Neural Information Processing Systems. 2025.
>
> [Pan et al, 2025]: Pan, Xu, et al. "Dissecting Query-Key Interaction in Vision Transformers." Advances in Neural Information Processing Systems 37 (2024): 54595-54631.
>
> [Merullo et al, 2024]: Merullo, Jack, Carsten Eickhoff, and Ellie Pavlick. "Talking heads: Understanding inter-layer communication in transformer language models." Advances in Neural Information Processing Systems 37 (2024): 61372-61418.

---

> > ### Author Rebuttal · Reviewer_4oH1 · 2026-04-03
> >
> > I am satisfied with their response and would maintain my positive score.

---

### Decision · Program_Chairs · 2026-04-30

**Decision:**

Accept (regular)

**Comment:**

This paper studies when and why singular vectors of attention heads align with model features, using a combination of toy models, theoretical analysis, and experiments on language models. Reviewers appreciated the combination of directly observable toy settings, formal analysis, and validation in real models, but several also raised concerns about novelty framing, the interpretation of the SAD control, and how clearly the real-model experiments support the main claim.

The main reservations concerned novelty and positioning. Several reviewers wanted the paper to be clearer that the contribution is not the discovery of alignment itself, but a theoretical and operational account of when such alignment should be expected and how it may be recognized. There was also lingering discomfort during the reviewer discussion phase about the use of low rank and how the authors run controls. I do not see it as a big blocker for technical soundness (people use effective ranks and ranks interchangeably sometimes in informal discussions) but the authors definitely need to improve the definitions to be clearer.

Overall, I view this as a borderline but positive paper and recommend weak accept. For the final version, the authors should tighten the novelty claim, clarify the matrix-rank/control discussion, and make the relationship to prior mechanistic-interpretability work more explicit.